# Genomic adaptation of the picoeukaryote *Pelagomonas calceolata* to iron-poor oceans revealed by a chromosome-scale genome sequence

Nina Guérin[1,2], Marta Ciccarella [1], Elisa Flamant[1,2], Paul Frémont [1,2], Sophie Mangenot[1,2], Benjamin Istace[1], Benjamin Noel [1], Caroline Belser [1], Laurie Bertrand[1,2], Karine Labadie [2,3], Corinne Cruaud[2,3], Sarah Romac [4], Charles Bachy [4,5], Martin Gachenot[5], Eric Pelletier[1,2], Adriana Alberti[1,2,6], Olivier Jaillon [1,2], Patrick Wincker [1,2], Jean-Marc Aury [1] & Quentin Carradec [1,2✉]

The smallest phytoplankton species are key actors in oceans biogeochemical cycling and their abundance and distribution are affected with global environmental changes. Among them, algae of the Pelagophyceae class encompass coastal species causative of harmful algal blooms while others are cosmopolitan and abundant. The lack of genomic reference in this lineage is a main limitation to study its ecological importance. Here, we analysed *Pelagomonas calceolata* relative abundance, ecological niche and potential for the adaptation in all oceans using a complete chromosome-scale assembled genome sequence. Our results show that *P. calceolata* is one of the most abundant eukaryotic species in the oceans with a relative abundance favoured by high temperature, low-light and iron-poor conditions. Climate change projections based on its relative abundance suggest an extension of the *P. calceolata* habitat toward the poles at the end of this century. Finally, we observed a specific gene repertoire and expression level variations potentially explaining its ecological success in low-iron and low-nitrate environments. Collectively, these findings reveal the ecological importance of *P. calceolata* and lay the foundation for a global scale analysis of the adaptation and acclimation strategies of this small phytoplankton in a changing environment.

[1] Génomique Métabolique, Genoscope, Institut François Jacob, CEA, CNRS, Univ Evry, Université Paris-Saclay, 91057 Evry, France. [2] Research Federation for the Study of Global Ocean Systems Ecology and Evolution, R2022/Tara Oceans GO-SEE, 3 rue Michel-Ange, 75016 Paris, France. [3] Genoscope, Institut François Jacob, CEA, Université Paris-Saclay, 2 Rue Gaston Crémieux, 91057 Evry, France. [4] Sorbonne Université, CNRS, Station Biologique de Roscoff, AD2M, UMR7144, Place Georges Teissier, 29680 Roscoff, France. [5] Sorbonne Université, CNRS, FR2424, Station Biologique de Roscoff, 29680 Roscoff, France. [6] Université Paris-Saclay, CEA, CNRS, Institute for Integrative Biology of the Cell (I2BC), 91198 Gif-sur-Yvette, France. ✉email: qcarrade@genoscope.cns.fr

Marine phytoplankton accounts for more than 45% of photosynthetic primary production on Earth and play an essential role in supplying organic matter to marine food webs[1]. They are key global actors in $CO_2$ uptake and provide gaseous oxygen to the atmosphere. A global decline of phytoplankton biomass has been reported over the past century (1% of chlorophyll-a concentration per year) leading to a decrease of net primary production in many oceanic regions[2]. This decline is probably a consequence of global ocean warming which drives water column stratification, reducing the nutrient supply to surface waters. Temperature-driven reductions in phytoplankton productivity in tropical and temperate regions are likely to have cascading effects on higher trophic levels and ecosystem functioning[3].

Photosynthetic picoeukaryotes (PPEs), defined by a cell diameter <3 µm, belong to different phyla, including Chlorophyta, Cryptophyta, Haptophyta, and Stramenopiles[4]. Present in all oceans, PPEs are the dominant primary producers in warm and oligotrophic regions[5]. Ocean warming and expansion of oligotrophic regions in the next decades may extend the ecological niche of PPEs, and a global shift from large photosynthetic organisms toward smaller primary producers is expected[3,6]. For example, sea ice melting in the Canadian Arctic Basin has been associated with an increase in the abundance of PPEs such as *Micromonas* at the expense of larger algae[7]. In the laboratory, this alga has the capacity to change its optimum temperature for growth in only a few hundred generations, suggesting that it will be less affected by global warming than many larger organisms[8]. In addition, the larger cell surface-to-volume ratio of PPEs compared to larger phytoplankton cells is advantageous for resource acquisition and growth in nutrient-limited environments[9,10].

Iron is one key compound required for the activity of the respiratory chain, photosynthesis and nitrogen fixation[10]. Because bioavailable iron is extremely low in more than one-third of the surface ocean, small phytoplankton has developed several strategies to optimize iron uptake and reduce iron needs[11]. In diatoms, reductive and non-reductive iron uptake mechanisms involve many proteins, including phytotransferrins, transmembrane ferric reductases, iron permeates, and siderophore-binding proteins[12]. The iron needs can be modulated by the variation of gene expression levels between iron-required proteins and their iron-free equivalent. These protein switches include electron transfer (flavodoxin/ferredoxin), gluconeogenesis (fructose-bisphosphate aldolase type I or type II) and superoxide dismutases (Mn/Fe-SOD, Cu/Zn-SOD or Ni-SOD)[13–15].

PPE growth is also limited by nitrogen (N) availability in large portions of the global ocean[16]. Ammonium ($NH_4^+$), nitrate ($NO_3^-$) and nitrite ($NO_2^-$) are the primary source of inorganic N for PPEs, however, several studies have shown that dissolved organic N, like urea, can be metabolized in N-limited environments[17]. For example, several membrane-localized urea transporters in the diatom *Phaeodactylum tricornutum* are maximally expressed in nitrogen-limited conditions[18] and the harmful algal blooms of the pelagophyceae *Aureococcus anophagefferens* may be fueled by urea[19].

Despite their large taxonomic distribution, most molecular studies on the ecological role of PPEs and their adaptation to the environment are restricted to a few species. PPEs are suspected of possessing highly developed acclimation/adaptation capacities, but the underlying molecular mechanisms remain poorly characterized due to the lack of reference genomic data.

Among PPEs, *Pelagomonas calceolata* was the first described member of the Pelagophyceae class[20]. It has since been identified in many oceanic regions using its 18 S rRNA sequence and chloroplast genome[21–23]. Several studies have demonstrated the capacity of *P. calceolata* to adapt to different environmental conditions. In the laboratory, *P. calceolata* has been shown to exhibit a high degree of

acclimation to light fluctuations with rapid activation of the photo-protective xanthophyll cycle and non-photochemical quenching[24]. In the Marquesas archipelago, *P. calceolata* is one of the most responsive species to iron fertilization with upregulation of genes involved in photosynthesis, amino acid synthesis and nitrogen assimilation[13]. A global-scale analysis of pelagophyte genes revealed that they are adapted to low-iron conditions[14]. In the subtropical Pacific, *P. calceolata* expresses stress genes in surface samples and genes involved in nitrogen assimilation are overexpressed in the deep-chlorophyll maximum[25]. A laboratory study suggests that *P. calceolata* also has the ability to increase the transcription levels of organic-nitrogenous compound cleavage enzymes (cathepsin, urease, arginase) under low nitrogen concentration[26]. Thus, gene expression appears to be controlled according to the nitrogen source and quantity. Taken together, this apparent adaptive plasticity may explain the presence of *P. calceolata* in many different oceanic environments, however, an exhaustive analysis of the genetic capacity of this species and the in situ characterization of its ecological niche is lacking.

Here we sequenced, assembled and annotated the *Pelagomonas calceolata* genome, with a combination of long- and short reads. We examined its genomic structure and gene content relative to other unicellular phytoplankton. We used this genome to detect *P. calceolata* in environmental datasets of *Tara* expeditions across all oceans to characterize its ecological niche and to identify the environmental conditions controlling its relative abundance. Finally, environmental expression levels of genes involved in nitrogen compounds and iron uptake and metabolism were studied.

## Results

**Chromosome-scale assembly and annotation of the *P. calceolata* genome.** To measure the abundance of *P. calceolata* in the oceans and study its genetic capacity to grow in different environmental conditions, we sequenced and assembled the genome of *P. calceolata* RCC100 using long reads of Oxford Nanopore Technologies (ONT) and Illumina short reads. Using the k-mer distribution of short reads, the genome was estimated to be homozygous with a size of 31 Mb (Supplementary Fig. S1a). The ONT long reads were assembled with Flye into six nuclear contigs for a total of 32.4 Mb, 1 plastid circular contig (90 Kb) and 1 mitochondrial circular contig (39 Kb) (Fig. 1, Supplementary Fig. S1b, c, and Supplementary Data 1). Two large and highly similar duplicated regions (>99% of identity) were detected at the extremity of contig 1 and 5 (393 Kb) and at the extremities of contig 3 and 6 (192 Kb; Supplementary Note 1 and Supplementary Fig. S1d). (TTAGGG)n telomeric repeats were detected at both ends of contigs 2, 3, 4, and 6, indicating that these four contigs represent complete chromosomes (Supplementary Fig. S1d). For contig 1 and contig 5, telomeric sequences were identified at only one extremity, the other extremity ending in the duplicated region. We used the Hi-C long-range technology to validate the assembly of *P. calceolata* genome sequence. The interaction map revealed a high number of contacts within contigs and very few across contigs (Supplementary Fig. S2 and Supplementary Note 2). No chimeras or fragmentations were detected. This result confirms that the six contigs correspond to six chromosomes of *P. calceolata*.

A total of 16,667 genes were predicted on the *P. calceolata* genome (see "Methods"), which is a high number for a PPE (Table 1 and Supplementary Data 2). There was an average of 0.45 intron per gene, and the distribution of their lengths reveals a peak at around 210 bp, which is the characteristic length of Introner Elements described in *A. anophagefferens*[27] (Supplementary Fig. S3, Supplementary Data 3, and Supplementary Note 3). In all, 9812 *P. calceolata* predicted proteins (58%) are homologous with at least

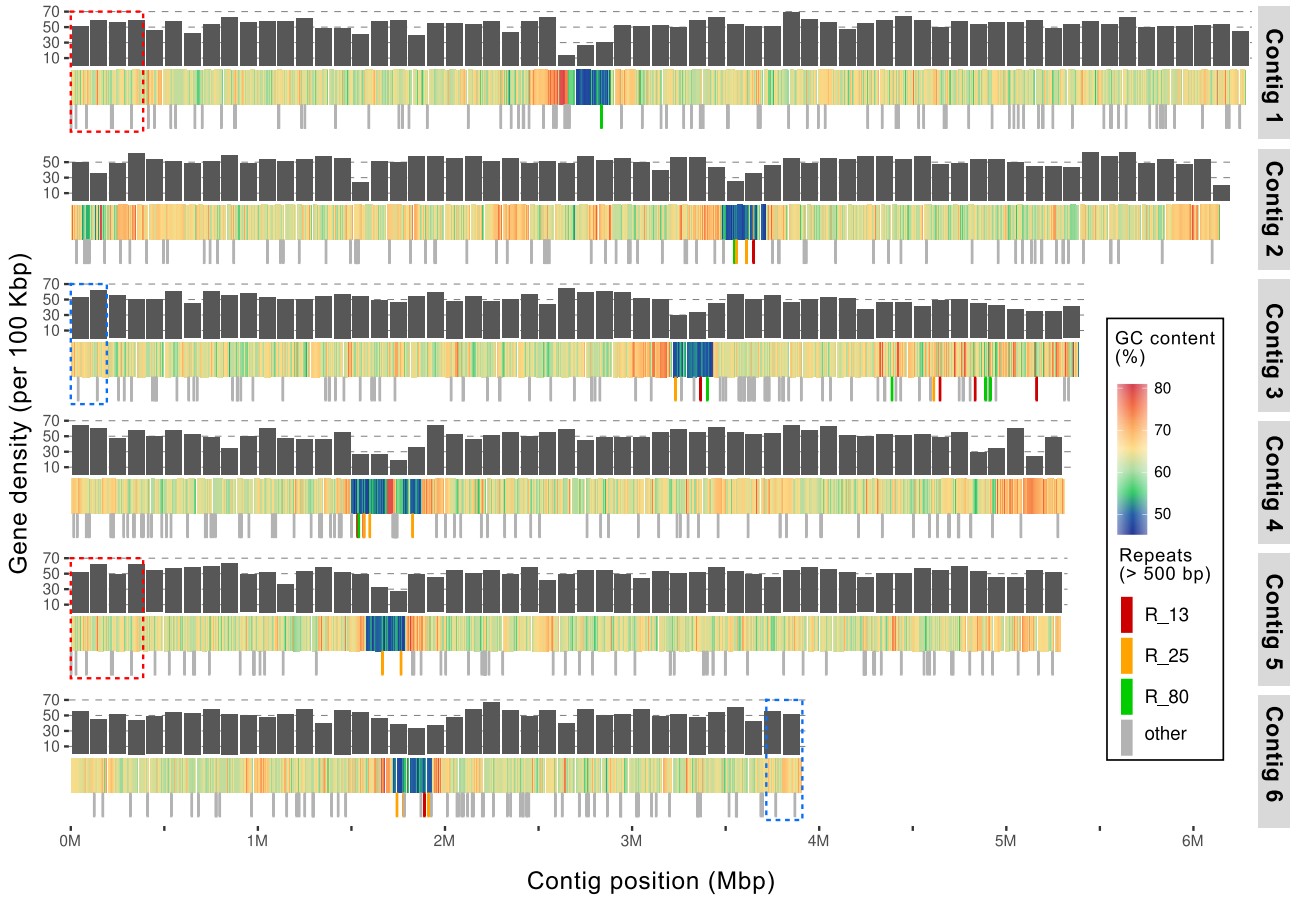

**Fig. 1 _Pelagomonas calceolata_ nuclear genome.** Representation of the 6 nuclear contigs of _P. calceolata_. The top layer indicates the number of genes per 100 Kb (black bars), the middle layer represents the GC content in percentage over a window of 200 Kb and the bottom layer is the position of DNA repeats of more than 500 bases repeated at least five times over the entire genome. Red, orange, and yellow bars indicate three different repeats in low-GC regions present in at least three different contigs. Dashed red and blue rectangles are duplicated chromosomic regions.

**Table 1 Genome characteristics of several unicellular photosynthetic eukaryotes.**

| Phylum/class | Species | Genome size (Mb) | Number of chromosomes | Predicted genes | GC% | Cell size | References |
|---|---|---|---|---|---|---|---|
| Pelagophyceae | _Pelagomonas calceolata_ | 32.4 | 6 | 16,667 | 63.6 | 2 μm | This study |
| Pelagophyceae | _Aureococcus anophagefferens_ | 56.0 | Unknown | 11,520 | 67.4 | 2 μm | 89 |
| Eustigmatophyceae | _Nannochloropsis oceanica_ | 29.3 | 32 | 7730 | 54.0 | 3 μm | 90 |
| Diatom | _Phaeodactylum tricornutum_ | 27.0 | 33 | 10,402 | 48.8 | 11 μm | 91 |
| Diatom | _Thalassiosira pseudonana_ | 34.5 | 24 | 11,776 | 46.9 | 5 μm | 92 |
| Chlorophyta | _Micromonas pusilla_ | 21.9 | 17 | 10,575 | 65 | ≤2 μm | 93 |
| Chlorophyta | _Ostreococcus lucimarinus_ | 13.2 | 21 | 7651 | 60 | 1.3 μm | 94 |
| Chlorophyta | _Bathycoccus prasinos_ | 15.1 | 19 | 7847 | 48 | 1–2 μm | 95 |
| Haptophyta | _Emiliania huxleyi_ | 141.7 | Unknown | 38,549 | 64.5 | 4–5 μm | 96 |

one gene in a stramenopile genome, including 2631 (16%) only shared with the pelagophyte _A. anophagefferens_ (Supplementary Fig. S4). A conserved functional domain (Pfam, KO or Inter-ProScan) was found in 11,240 proteins (67%). Even if gene completeness estimations are imprecise for species distant from model organisms, we obtained 94.0% of completeness with BUSCO[28] (88% single-copy and 6% duplicated genes), showing that our genome is near complete (Supplementary Data 4).

**GC content, centromeres, and meiosis in _P. calceolata_.** A remarkable feature in the _P. calceolata_ genome is the distribution of GC content along _P. calceolata_ chromosomes (Fig. 1). While the average GC content of the nuclear genome is 63%, one large region

in each contig (259 Kb on average) is 52% GC. These unique large troughs in GC content in each chromosome suggest that these regions encompass centromeres. The Hi-C result confirms centromere positions with the presence of contacts between low-GC regions across chromosomes, suggesting the physical proximity of these regions in the nucleus (Supplementary Fig. S2). Interestingly, we did not observe an accumulation of repeated elements or transposons in these low-GC regions and only a slight decrease of gene density. Genomic specificities and gene content of low-GC regions are detailed in Supplementary Data 5, S6 and Supplementary Note 4. Low-GC patterns in centromeres could be explained by the inhibition of recombination[29,30], suggesting that _P. calceolata_ is capable of meiosis and recombination. Among 23 meiosis-specific

genes characterized in other species[31–33], 18 homologs are present in the *P. calceolata* genome (Supplementary Data 7). These genes include the double-strand DNA break (DSB) initiator *SPO11*; *RAD50*, *RAD52,* and *MRE11* to bind DSBs; *HOP2*, *MND1*, *DMC1*, and *RAD51* to ensure pairing and invade the homologous strand; *MSH2*, *MSH3 PMS1*, and *MSH6* genes involved in the synthesis-dependent strand annealing pathway and *MUS81* necessary for non-interfering (class II) crossing over. The five missing genes are absent in many organisms capable of meiosis, suggesting that they are not required to perform genetic recombination (Supplementary Note 5). Taken together, the genomic structure and the genetic content of *P.calceolata* strongly suggests that this species performs meiosis.

**Relative abundance of *P. calceolata* across oceanic basins.** To estimate the relative abundance of *P. calceolata* across all oceans, we first used the abundance of the V9 region of the 18 S rRNA sequenced from all samples of the *Tara* Oceans expedition[34]. The most abundant *P. calceolata* Operational Taxonomic Unit (OTU) in the 0.8–5 µm size fraction is on average 0.80% for the 104 surface samples and 1.23% for the 61 deep-chlorophyll maximum (DCM) samples (Supplementary Data 8). According to this abundance estimation method, *P. calceolata* is the third most abundant eukaryote OTU of the 0.8–5 µm size fraction after two Dinophyceae OTUs affiliated to *Ankistrodinium* and an unknown Gymnodiniaceae.

To estimate *P. calceolata* abundance independently from PCRs and 18 S rRNA copy number bias, we used the mapping of metagenomic reads on the *P. calceolata* genome. For the 0.8–5 µm size fraction, the percentage of sequenced reads aligned on the genome is 1.39% ($n = 93$, sd = 1.5) in surface samples and 2.67% ($n = 55$, sd = 1.6) in DCM samples. In the 0.8–2000 µm size fraction, *P. calceolata* represents 1.01% ($n = 80$, sd = 1.2) of all reads in surface samples and 1.56% ($n = 39$, sd = 1.3) in DCM samples. A maximal relative abundance of 6.7% in the $0.8 − 5$ µm size fraction was observed in the North Indian Ocean (station TARA_38) at the DCM (Fig. 2a). In the Indian Ocean, Red Sea and Mediterranean Sea, *P. calceolata* is significantly more abundant in the DCM than at the surface (Fig. 2b). In cold waters (below 10 °C), *P. calceolata* is not detected above our threshold of 25% of horizontal genomic coverage. Important variations between and within each oceanic basin are observed, suggesting that many biotic or abiotic factors influence *P. calceolata* abundance.

Finally, we compared the two methods of abundance estimations (Supplementary Fig. S5). The metagenomic-based relative abundance is strongly correlated to the metabarcoding-based relative abundance (Pearson correlations of 0.91 and 0.70 for the 0.8–2000 and 0.8–5 µm size fractions, respectively). However, the metabarcoding-based abundance is on average 2.3 lower in the 0.8–5 µm size fraction and 3.1 lower in the 0.8–2000 size fraction compared to the metagenomic-based abundance (Supplementary Fig. S5).

**High relative abundance of *P. calceolata* in temperate, low-light, and iron-poor regions.** In order to identify factors controlling *P. calceolata* abundance in the oceans, we used physical-chemical parameters available for each oceanic station (see "Methods"). Principal component analysis revealed a positive relation between metagenomic-based *P. calceolata* abundance, the temperature, and the coast distance and a negative relation with iron concentration (Fig. 3a, b). This result was consistent over the 2 size fractions containing *P. calceolata* cells (0.8–5 µm and 0.8–2000 µm size fractions; Supplementary Fig. S6). Despite the numerous factors potentially influencing *P. calceolata* abundance, we observed a weak but significant Pearson's positive correlation with the

temperature, a negative correlation with Photosynthetically Active Radiation (PAR, mean of 30 days) and a negative correlation with iron concentrations (Table 2). In the 0.8–5 µm size fraction, the relative abundance of *P. calceolata* is higher in low-iron conditions (<0.2 nmol/l, 54 samples) with on average 2.3% of metagenomic reads than in high-iron environments (>0.2 nmol/l, 88 samples) with on average 1.7% of metagenomic reads (Wilcoxon test, *P* value = 0.02). In the 0.8–2000 µm size fraction, we observe the same tendency with a relative abundance of 1.9% of metagenomic reads on average in low-iron waters (49 samples) and a lower relative abundance of 0.78% of metagenomic reads on average in high-iron environments (59 samples) (Wilcoxon test, *P* value = 9.6e$^{-7}$). In addition, *P. calceolata* relative abundance is weakly correlated with the 9'butanoyloxyfucoxanthin concentration, a signature pigment for pelagophytes (Pearson 0.22, *P* value = 0.02 and Pearson 0.41, *P* value = 4.82e$^{-05}$ in the 0.8–5 µm and 0.8–2000 µm size fraction, respectively). We used a general additive model to estimate the contribution of temperature, PAR and iron concentration to *P. calceolata* relative abundance (Table 2). The three factors explain 32.3% of the variations of *P. calceolata* abundance in the 0.8–5 µm size fraction and 56.8% in the 0.8–2000 µm size fraction.

Finally, we projected the relative abundance of *P. calceolata* at the end of the century following Frémont et al. methodology[35]. We modeled the ecological niche of *P. calceolata* using the World Ocean Atlas (WOA18) datasets at the time and location of sampling or using the projected climatology in 2099 using the RPC8.5 scenario (see "Methods"). We used four machine-learning techniques: Generalized Additive Models (GAM), Neural Networks (nn), Random Forest (rf) and Gradient Boosted Trees (bt) and evaluated their performances with two parameters. The Pearson correlation coefficient indicates the correlation between the model and in situ measurements of *P. calceolata* abundance. The four machine-learning tools have similar performances based on Pearson's correlations (nn = 0.676; gam = 0.621; bt = 0.683; rf = 0.694). The second parameter is the root mean square error (rmse) and reflects the magnitude of the errors in the models (the number of standard deviations from the mean). Using this metric the GAM approach is less good (rmse = 1.04) than the three other tools (nn = 0.964; bt = 0.952; rf = 0.941). These results indicate that we have enough in situ data to capture the global trends on the relative abundance of *P. calceolata* but these models could be imprecise on the amplitude of abundance variations. In addition, the predictions in the tropical waters are uncertain because this environment in 2099 is out of the range of the training dataset. Because the performances of the four models are similar, we combined them to obtain the most accurate projection (Fig. 3c and Supplementary Fig. S7). Despite these limitations, we projected an increase of up to 1.12% of *P. calceolata* relative abundance from latitude 40° to latitude 50° in the North and South hemispheres and a decrease in temperate and tropical waters (−0.8% maximum).

**Genes related to iron uptake, storage and usage in *P. calceolata*.** Iron is a critical metal for all photosynthetic organisms, required for the photosynthesis, the nitrogen cycle and the protection against reactive oxygen species. Since *P. calceolata* seems to thrive in iron-poor waters, we identified *P. calceolata* genes coding for iron uptake and storage, then compared their expression levels in low (<0.2 nmol/l) versus high (>0.2 nmol/l) iron conditions using *Tara* Oceans metatranscriptomes. *P. calceolata* has five genes encoding the phytotransferrin *ISIP2A* involved in Fe$^{3+}$ uptake via endosomal vesicles and 2 putative iron-storage protein *ISIP3*. In iron-poor environments, three *ISIP2A* and one *ISIP3* are over-expressed (Fig. 4a and Supplementary Data 9). This result indicates

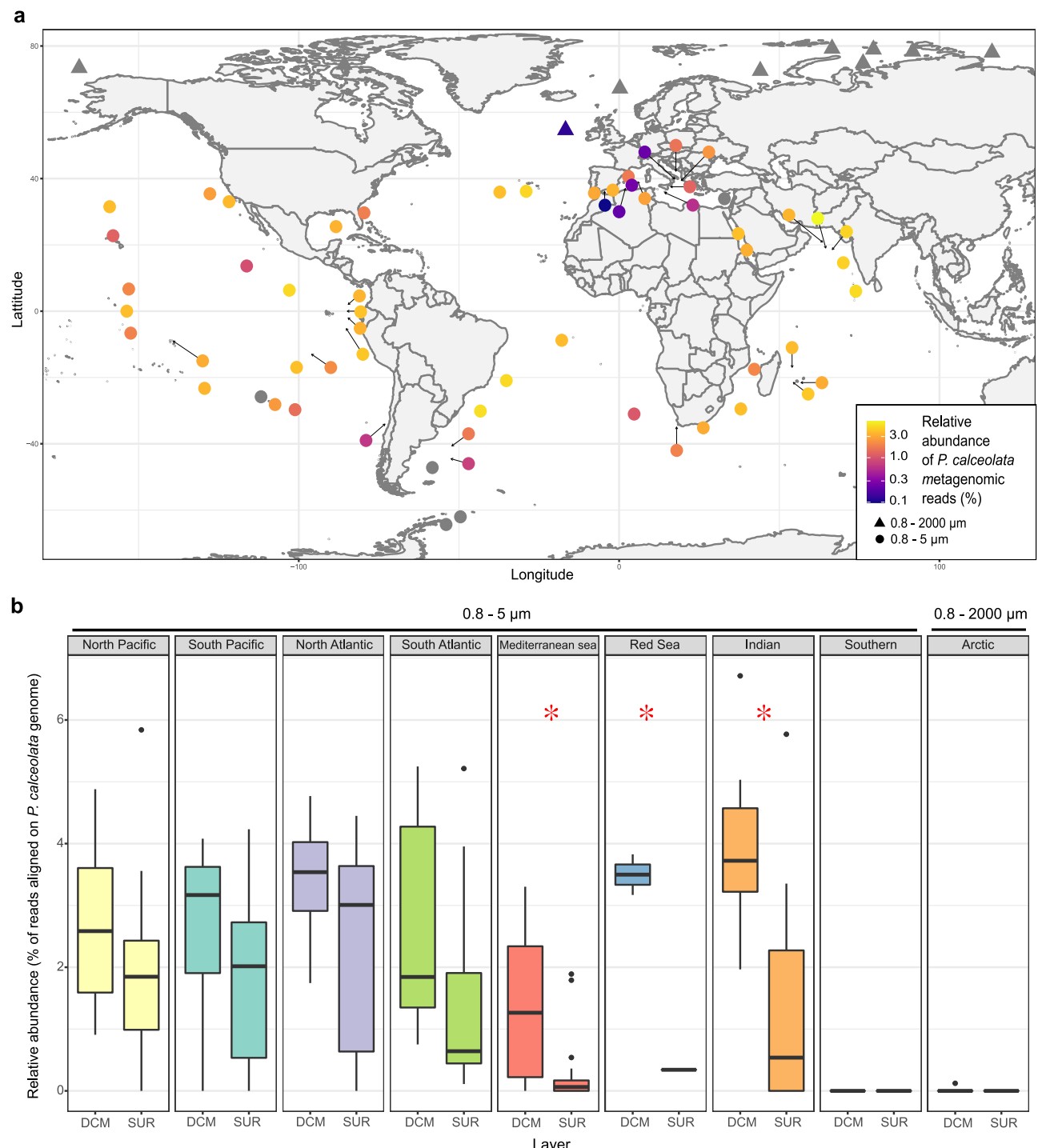

**Fig. 2 Relative abundance and distribution of _P. calceolata_ in the oceans. a** World map of the relative abundance of _P. calceolata_ metagenomic reads. The color code indicates the percentage of sequenced reads aligned on the genome. The DCM samples of size fractions 0.8–5 μm (circles) or 0.8–2000 μm (triangles) are shown. _P. calceolata_ is considered to be absent when the horizontal coverage is below 25% of the genome (gray dots). **b** Boxplot of the relative abundance in each oceanic region in surface and DCM samples. Red stars indicate a significant difference between SUR and DCM samples (Wilcoxon test, _P_ value <0.01).

that similarly to diatoms, ISIP are upregulated following iron starvation in _P. calceolata_ potentially improving cell growth in low-iron environments. Three genes encode the iron transporter ferroportin but are not differentially expressed according to the environment (Supplementary Fig. S8a). These proteins are iron exporters in multicellular organisms but their function in microalgae remains to be studied[36]. Finally, we identified eight Zinc/iron permeases potentially involved in iron uptake from the

environment in the _P. calceolata_ genome. Among them, two are overexpressed in high-iron and one in low-iron environments. Interestingly, we note the absence of the iron permease _FTR1_, the iron-storage ferritin and the starvation induced protein _ISIP1_ (involved in endocytosis of siderophores in diatoms). In comparison to _P. calceolata_, the coastal Pelagophyceae _A. anophagefferens_ do not have _ISIP3_ gene and a lower number of Zinc–Iron permeases.

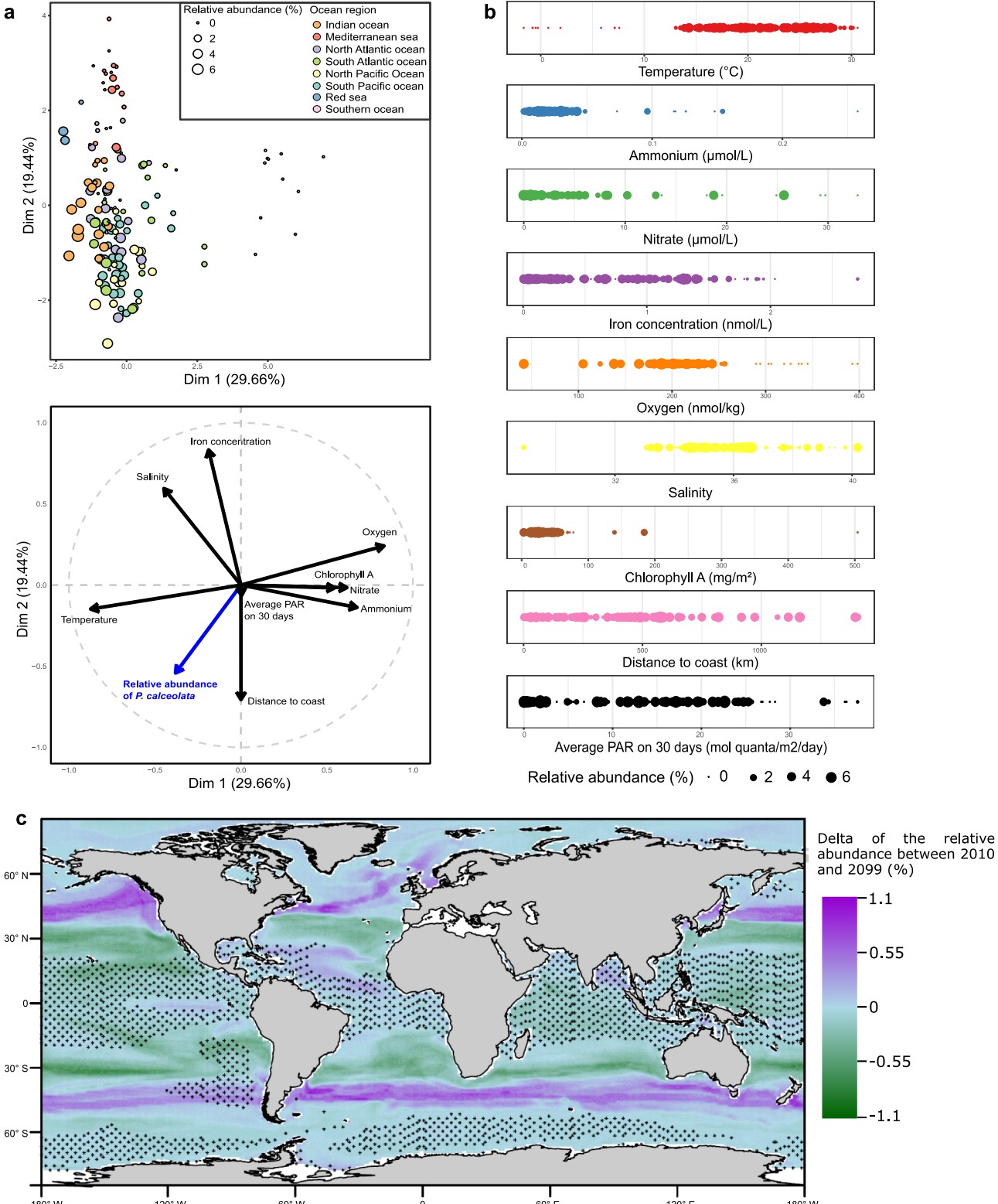

**Fig. 3 Ecological niche of *P. calceolata*. a** Principal component analysis of the metagenomic-based relative abundance of *P. calceolata* in the 0.8–5 µm size fraction. Percentages of variance explained by each axis are indicated on axis titles. Top panel: each dot represents a sample with a size proportional to the relative abundance of *P. calceolata* and the colors indicate the oceanic basins. Bottom panel: nine environmental parameters are represented as vectors alongside the relative abundance of *P. calceolata* (blue vector). **b** Bubble plot of the relative abundance of *P. calceolata* for the 0.8–5 µm size fraction according to the nine environmental parameters. **c** Delta of the modeled relative abundance of *P. calceolata* between 2010 and 2099. Green areas correspond to a decrease while purple areas correspond to an increase of *P. calceolata* relative abundance. Small stars indicate locations where at least one of the predictor drivers is out of range compared to the training dataset values.

**Table 2 Environmental parameters explaining *P. calceolata* relative abundance for the 0.8–5 µm (a) and the 0.8–2000 µm (b) size fractions.**

| (a) | GAM model | | | GAM verification | | Pearson correlations | |
|---|---|---|---|---|---|---|---|
| 0.8–5 µm | edf | F value | P value | k-index | k P value | r | P value |
| s(Temperature) | 1 | 22.16 | 6.84e$^{-6}$ | 1.11 | 0.87 | 0.23 | 0.001 |
| s(Iron concentration) | 1.257 | 13.12 | 2.26e$^{-4}$ | 0.93 | 0.12 | −0.25 | 0.001 |
| s(PAR 30 days) | 1.859 | 15.94 | 4.56e$^{-7}$ | 1.01 | 0.44 | −0.32 | 0.001 |
| Adjusted $R^2$ | 0.3 | | | | | | |
| Deviance explained | 32.30% | | | | | | |

| (b) | GAM model | | | GAM verification | | Pearson correlations | |
|---|---|---|---|---|---|---|---|
| 0.8–2000 µm | edf | F value | P value | k-index | k P value | r | P value |
| s(Temperature) | 3.628 | 9.442 | 1.71e$^{-6}$ | 0.96 | 0.31 | 0.57 | 0.0001 |
| s(Iron concentration) | 1.54 | 1.225 | 2.42e$^{-1}$ | 0.91 | 0.14 | −0.47 | 0.0001 |
| s(PAR 30 days) | 2.454 | 6.962 | 0.00027 | 1.14 | 0.93 | −0.051 | 0.6 |
| Adjusted $R^2$ | 0.53 | | | | | | |
| Deviance explained | 56.80% | | | | | | |

Several important ferrous proteins can be substituted by non-ferrous equivalents in iron-poor environments[14]. In the *P. calceolata* genome we identified 11 flavodoxin genes involved in electron transfer during photosynthesis, potentially replacing ferredoxins (17 genes; Supplementary Data 9). This number of genes is important compared to other algae, including *A. anophagefferens* (5 flavodoxins and 9 ferredoxins). Expression levels of these genes across the oceans revealed overexpression of flavodoxin genes in low-iron environments, replaced by ferredoxins in high-iron conditions (Fig. 4b). The Fructose-Bisphosphate Aldolase (FBA), necessary for gluconeogenesis and the Calvin cycle, is encoded by six genes in *P. calceolata*. Two genes are dependent on a divalent cation (FBA type II), and the four others are Zinc/Iron-independent (FBA type I). FBA type I is overexpressed in high-iron conditions and quasi-absent in low-iron conditions (Fig. 4c). Finally, all types of Superoxide dismutases (SOD) are found in the *P. calceolata* genome. Non-ferrous SODs (Cu/Zn and Ni) encoded by three genes and Mn/Fe-SOD encoded by two genes are not differentially expressed according to iron concentrations (Supplementary Fig. S8b). The gene content and the transcriptomic flexibility suggest important capacities for *P. calceolata* growth in low-iron environments.

**Genes involved in nitrogen uptake, storage, and recycling.** Because *P. calceolata* could be an important player in the nitrogen (N) cycle in oceanic ecosystems[25], we explored its gene content and analyzed the expression levels of genes involved in nitrogen metabolism in the environment using *Tara* Oceans metatranscriptomes (Fig. 5 and Supplementary Data 9). The uptake of nitrogen-containing inorganic compounds is supported by 11 genes in *P. calceolata*. Three genes encode nitrate/nitrite transporters, three genes encode formate/nitrite transporters and five genes encode ammonium transporters. One formate/nitrite and two nitrate/nitrite transporter genes are significantly overexpressed in low-nitrate conditions (Supplementary Fig. S9). One ammonium transporter is overexpressed in low-ammonium environments (Supplementary Fig. S9). We identified one nitrate reductase and two nitrite reductases in the *P. calceolata* genome (Supplementary Data 9). The expression level is significantly higher in high-nitrate environments for the nitrate reductase and one nitrite reductase (Supplementary Fig. S10). But the second nitrite reductase is surprisingly less expressed in high-nitrate conditions. The number of enzymes incorporating ammonium into organic compounds (GS/GOGAT pathway) is higher in *P. calceolata* than in other species: five glutamine synthetase (GS) and four glutamate synthase (GOGAT) genes are present in the *P. calceolata* genome. Two GS genes are more expressed in high-nitrate samples and 1 GOGAT gene is more expressed in low-nitrate samples (Supplementary Fig. S10).

We identified three genes carrying the nitrate and nitrite sensing (NIT) domain (IPR013587) in the *P. calceolata* genome. Using NCBI non-redundant proteins and marine genomic databases (see "Methods"), 60 homologous proteins of the NIT-sensing domains of *P. calceolata* were identified. These homologs are restricted to the Pelagophyceae class (16 transcriptomes), the Dictyochophyceae class (6 transcriptomes) and one putative cryptophyte transcriptome. The phylogenetic tree of this protein family shows three subfamilies diverging before the Dictyochophyceae/Pelagophyceae separation (Supplementary Fig. S11). One *P. calceolata* protein carries a NIT-sensing domain surrounded by two transmembrane domains suggesting a capacity for external nitrate/nitrite sensing while the two other NIT genes carry a protein-kinase domain (IPR000719) suggesting phosphorylation-based signal transduction dependent on intracellular nitrate or nitrite concentration (Fig. 5b). To investigate this possibility, we studied the expression levels of NIT-sensing genes. The potential external NIT-sensing gene is indeed significantly overexpressed in low-nitrate environments. In contrast, only one of the intracellular NIT-sensing genes is upregulated in nitrate-rich environments (Fig. 5c). This result suggests an important role of the NIT-sensing genes in the acclimation to environmental nitrate concentrations.

Finally, we identified genes involved in nitrogen recycling from organic compounds which are important in several species in case of inorganic nitrogen deprivation. One arginase and one cyanate lyase were detected in the *P. calceolata* genome but no gene encoding formamidase. In addition, the number of gene copies for enzymes involved in the urea cycle (carbamoyl-phosphate synthetase, ornithine carbamoyltransferase, argininosuccinate synthase and argininosuccinate lyase) is equal or slightly lower than in other algae (Fig. 5a and Supplementary Data 9). Among these genes, only the cyanate lyase is overexpressed in low-nitrate conditions suggesting cyanate is an important alternative source of organic nitrogen for *P. calceolata* (Supplementary Fig. S10).

**Discussion**
The essential roles of phytoplankton in oceanic ecosystems have been illustrated many times, however, numerous lineages are still poorly explored and model organisms are restricted to a few taxa (mainly diatoms, prasinophytes, and haptophytes) limiting the global understanding of phytoplankton activity. The *P. calceolata*

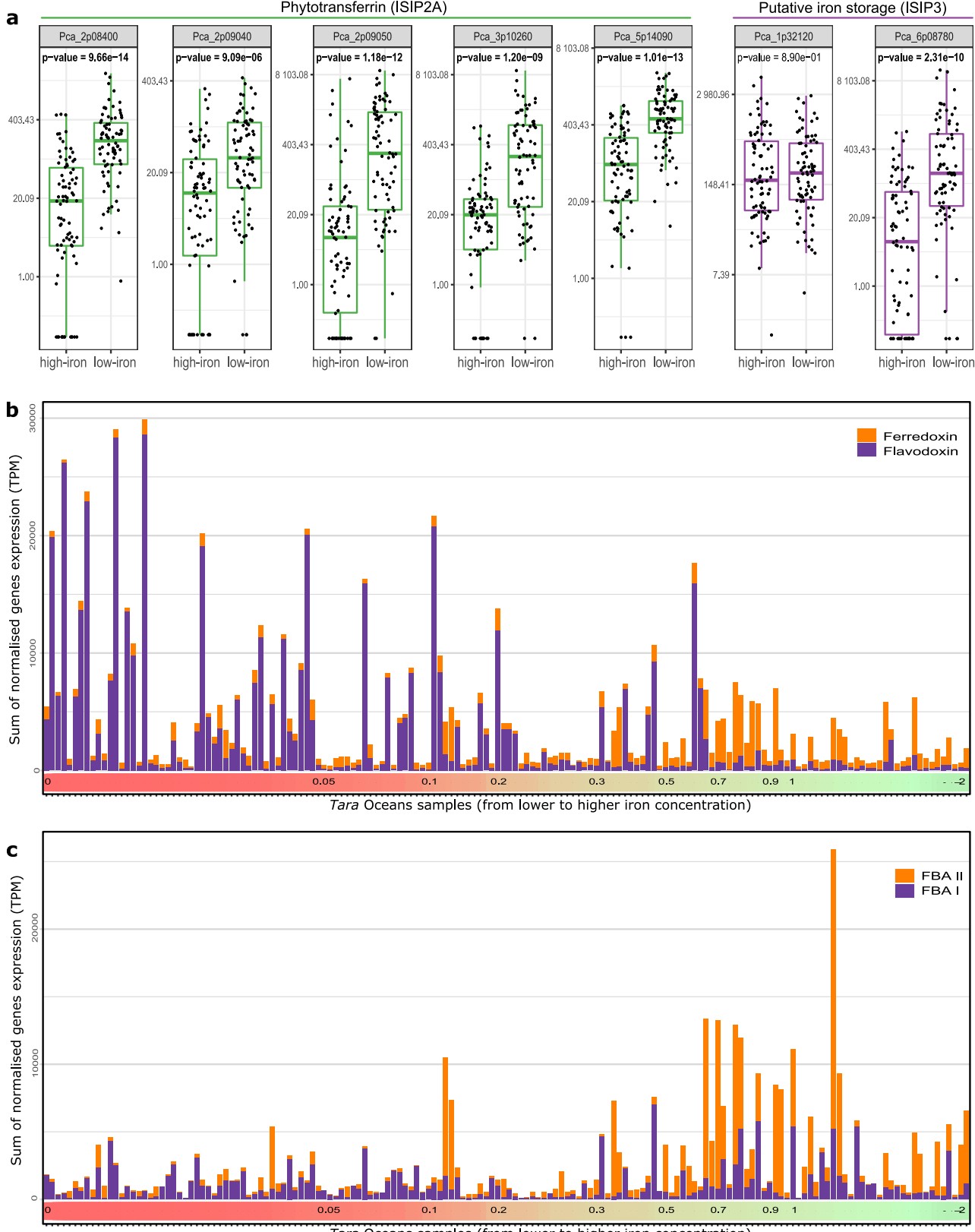

**Fig. 4 Expression of iron-related genes in *P. calceolata*. a** Relative gene expression levels normalized in transcript per million (TPM) of five phytotransferrins (*ISIP2A*) and two putative iron storage (*ISIP3*) in low-iron (<0.2 nM) and high-iron (>0.2 nM) oceanic stations. *P* values of Wilcoxon statistical tests between low- and high-iron conditions are indicated for each gene. Significant *P* values (<0.01) are in bold. **b** Relative expression levels (TPM) of genes coding for ferredoxin (orange) and its non-ferrous equivalent flavodoxin (purple) in each *Tara* Oceans sample. Samples are sorted from low-iron (left) to high-iron (right) conditions. Iron concentrations are indicated in nM on the colored horizontal bar. **c** Same representation for genes coding for fructose-bisphosphate aldolase II (orange) and its non-ferrous equivalent fructose-bisphosphate aldolase I (purple).

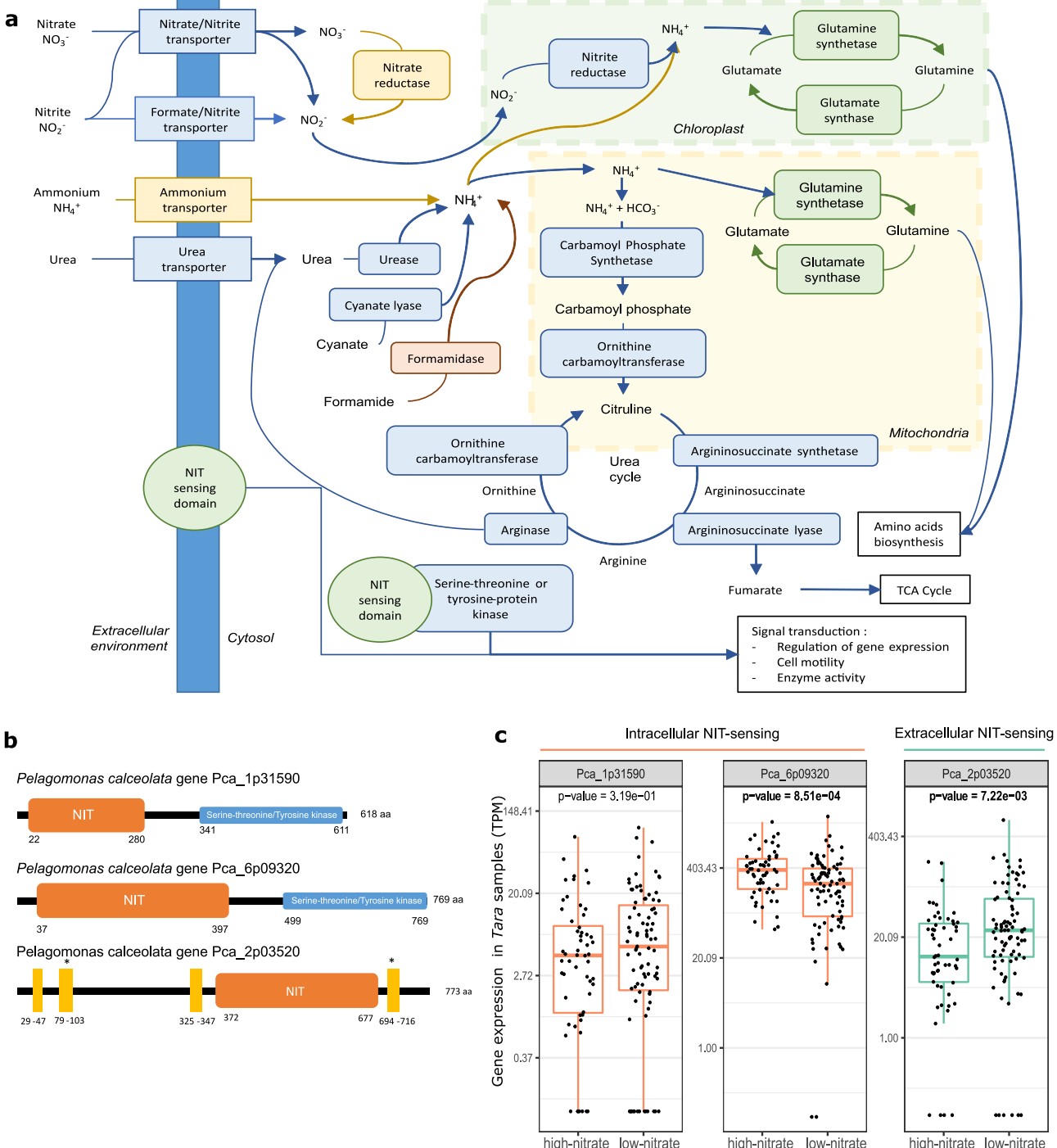

**Fig. 5 Nitrogen sensing and metabolism in *P. calceolata*. a** Schematic representation of N transport and assimilation in *P. calceolata* based on the gene content. The color code indicates if the number of gene copies for a specific function is overrepresented (green), equally represented (blue), underrepresented (orange) or absent (red) in *P. calceolata* genome compared to the mean of eight pico-nano photosynthetic eukaryotes. Gene copy number for each function is indicated in Supplementary Data 9. **b** Domain organization of NIT-sensing proteins in *P. calceolata*. Orange boxes are NIT-sensing domains (IPR13587), blue boxes are serine–threonine/tyrosine-kinase domain (IPR20635), and yellow rectangles are transmembrane domains. **c** Relative expression levels (TPM) of three NIT-sensing genes in low-nitrate (<2 μM) and high-nitrate (>2 μM) environments. *P* values of Mann–Withney–Wilcoxon tests between low- and high-nitrate samples are indicated for each gene. Significant *P* values (<0.01) are in bold.

genome assembled and annotated in this study reveals a previously underestimated high abundance of *P. calceolata* in the oceans and brings new insights into specific genomic features of this algae class related to its adaptation to specific environments.

We have shown in this study that *P. calceolata* is cosmopolitan in oceanic samples above 10 °C with a relative abundance

generally >1% of all sequenced reads. In contrast to the coastal Pelagophyceae *A. anophagefferens* that can present high peaks of abundance[37], no *P. calceolata* blooms were reported, but *P. calceolata* is well-adapted to an extensive range of environmental conditions as suggested by previous studies[21,23]. Although the abundance of an organism calculated from metabarcoding or

metagenomic data provides only an indirect and relative quantification of organism abundances, both methods suggest that *P. calceolata* is one of the most abundant pico-nano eukaryote in offshore data. The high relative abundance of *P. calceolata* measured with a metabarcoding approach has recently been confirmed with a qPCR method (average of 5 882 ± 2 855 rRNA gene copies mL$^{-1}$ on the surface of the eastern North Pacific)[21]. In addition, we have shown that the metabarcoding approach probably underestimates the relative abundance of *P. calceolata* compared to the metagenomic analysis owing to the low copy number of rRNAs in this organism. However, we cannot exclude that the large genome size of *P. calceolata* compared to bacterial genomes present in the 0.8–5 μm size fraction overestimates its relative abundance in metagenomic datasets. Further studies may combine microscopic and flow sorting approaches with genomic data to assess the number of cells and the biomass of this organism in the oceans. Our model analysis has revealed a probable increase of *P. calceolata* relative abundance at the end of the century in high latitudes where the seawater temperature is currently too low for this species. This result is coherent with previous studies suggesting a global increase of phytoplankton in subpolar regions[38,39].

Iron is essential for growth, photosynthesis, primary production, nitrogen fixation and reduction for PPEs[40]. Our results show that *P. calceolata* thrives in iron-poor waters and thus occupies a large ecological niche for a PPE. Two main strategies exist against iron deprivation: the optimization of iron uptake and the modulation of iron needs. Genes coding for iron chelators and ferritin are absent from the *P. calceolata* genome, and genes coding for passive iron transporters are under- or equally represented compared to other PPEs. In contrast, phytotransferrins (*ISIP2A*), putative iron-storage proteins (*ISIP3*) and ferroportins are over-represented in the *P. calceolata* genome. The expression levels of ISIP genes are anti-correlated with iron concentration in *P. calceolata* showing an acclimation to low-iron conditions. Because phytotransferrins are dependent on carbonate ions, ocean acidification may reduce iron uptake efficiency in many species including *P. calceolata*[41,42]. Compared to *P. calceolata*, the low abundance of *A. anophagefferens* in open oceans where iron is limited is consistent with the absence of genes involved in iron uptake and storage, including iron permeases and *ISIP3* genes. The presence of three ferroportin genes in *P. calceolata* is interesting since these transmembrane iron export proteins play a significant role in iron homeostasis in multicellular organisms[36]. Ferroportin function in microalgae is unknown but could act to export iron from endosomes to the cytoplasm[43]. In the green alga *Chlamydomonas reinhardtii*, a ferroportin gene is overexpressed under low Fe conditions[44]. Even though we did not find significant changes in the expression of the 3 ferroportin genes of *P. calceolata* according to iron concentration, the function of these genes could be investigated to understand their role in variable iron concentrations.

Modulation of iron needs seems to be a major acclimation strategy for *P. calceolata* in low-iron environments. All known molecular switches between ferrous and non-ferrous proteins are present in the *P. calceolata* genome and the transcriptomic regulation of these genes suggests a central role of these proteins for its growth in low-iron environments.

Expressing more than 90% of all nitrate transporter transcripts, pelagophytes may dominate nitrate uptake and assimilation in the North Pacific Ocean[25]. Indeed, *P. calceolata* contains a large collection of genes for inorganic nitrogen transporters. The main difference with the coastal *A. anophagefferens* is the reduced number of ammonium transporters in *P. calceolata*.

Organic nitrogen compounds could also be a major nitrogen source of for *P. calceolata*. We have shown that the cyanate lyase and urease genes are expressed in many environments but only the cyanate lyase is overexpressed in low-nitrate conditions. These two genes, largely present among phytoplankton lineages, could be significant components of acclimation to low-nitrate environments[45]. In addition, *A. anophagefferens* grow faster on the organic nitrogen substrates (urea and glutamic acid) than on nitrate or ammonium, suggesting that the dominance of Pelagophyceae in low-nitrate environments could be due to an optimized usage of these organic molecules[46].

One remarkable feature of the *P. calceolata* genome is the presence of three genes carrying NIT domains (PF08376). This NIT domain was first described in bacterial nitrite and nitrate sensor proteins[47]. This sensor is an alpha-helical protein playing a signal transduction role in regulating gene expression, cell motility and enzyme activity in *Klebsiella oxytoca*[48]. In pico-nano algae, NIT-sensing domains can be associated with a serine–threonine/tyrosine-kinase domain, suggesting signal transduction according to intracellular nitrate/nitrite concentration, or surrounded by two transmembrane domains suggesting extracellular sensing. Even though NIT-sensing domains can be found across various phyla, homologs of *P. calceolata* NIT proteins are restricted to pelagophytes and dictyochophytes. The NIT genes in *P. calceolata* are highly expressed in subtropical Pacific N-depleted waters, suggesting that these proteins have a role in transcription regulation according to nitrate availability[25]. Our results suggest that NIT-sensing proteins respond differently to environmental nitrate depletion. We can hypothesize that the NIT-sensing protein overexpressed in nitrate-rich environments plays a role in the intracellular regulation of stored nitrogen, activating pathways when nitrate or nitrite stocks are sufficient. In contrast, putative extracellular NIT-sensing could be an environmental nitrate or nitrite sensor activated to regulate the expression of genes involved in the acclimation to low-nitrate conditions.

In summary, due to its widespread distribution and its high abundance in the open oceans, *Pelagomonas calceolata* can serve as an ecologically-relevant model to study marine photosynthetic protists. We used the chromosome-scale genome sequence, mostly telomere-to-telomere, generated in this study to estimate its abundance in environmental datasets. We have shown that the *P. calceolata* genome has specific genomic features potentially explaining its ecological success in open oceans. Compared to the coastal Pelagophyceae *A. anophagefferens*, the *P. calceolata* genome contains specific genes involved in the acclimation to low-iron conditions. The large repertoire of genes involved in nutrient acquisition from the environment is coherent with its widespread pattern of relative abundance distribution across different environments. The ecological niche of *P. calceolata* suggests that this alga will benefit from the global climate change with the extension of oligotrophic regions and global ocean warming. Future studies could use the *P. calceolata* genome to explore adaptation and acclimation processes controlling the distribution and abundance of this alga.

## Methods

**Pelagomonas culture**. *Pelagomonas calceolata* RCC100 culture was grown in 12:12-h light:dark photoperiod in K medium with natural seawater base at 20 °C. At the Roscoff Culture Collection, cells were kept at a light intensity of ~80 μmol photon m$^{-2}$ s$^{-1}$ and the volume of culture was ramped up to 1 litre in mid-exponential growth phase before harvesting. RCC100 culture was not axenic and grown in the presence of undefined bacterial microbiota.

**DNA extraction, library preparation, and sequencing**. We pelleted cells from 500 ml of culture by two successive centrifugations at 10,000 × g for 15 min at 4 °C. Genomic DNA was extracted using the NucleoSpin Plant II Mini kit according to the manufacturer's instructions (Macherey-Nagel, Hoerdt, France) with the following exception for the lysis step: 400 μL of lysis buffer PL1 and 25 μL of proteinase K 25 mg/mL were added to strain pellets, and lysates were incubated at 55 °C for 1 h at 900 rpm. DNA quantity and integrity were respectively evaluated

on a Qubit 2.0 spectrofluorometer (Invitrogen, Carlsbad, CA, USA) and a Nanodrop1000 spectrophotometer (Thermo Fisher Scientific, MA, USA).

For Illumina sequencing, DNA (1.5 µg) was sonicated using a Covaris E220 sonicator (Covaris, Woburn, MA, USA). Fragments were end-repaired, 3′-adenylated and Illumina adapters (Bioo Scientific, Austin, TX, USA) added using the Kapa Hyper Prep Kit (KapaBiosystems, Wilmington, MA, USA). Ligation products were purified with AMPure XP beads (Beckmann Coulter Genomics, Danvers, MA, USA). The library was then quantified by qPCR using the KAPA Library Quantification Kit for Illumina Libraries (KapaBiosystems), and the library profile was assessed using a High Sensitivity DNA kit on an Agilent Bioanalyzer (Agilent Technologies, Santa Clara, CA, USA). The library was sequenced on an Illumina NovaSeq instrument (Illumina, San Diego, CA, USA) using 150 base-length read chemistry in paired-end mode.

For ONT sequencing, the library was prepared using the 1D Native barcoding genomic DNA (with EXP-NBD104 and SQK-LSK109). Genomic DNA fragments (1 µg) were repaired and 3′-adenylated with the NEBNext FFPE DNA Repair Mix, and the NEBNext® Ultra™ II End Repair/dA-Tailing Module (New England Biolabs, Ipswich, MA, USA). Adapters with barcodes provided by ONT were then ligated using the NEB Blunt/TA Ligase Master Mix (NEB). After purification with AMPure XP beads (Beckmann Coulter, Brea, CA, USA), the sequencing adapters (ONT) were added using the NEBNext Quick T4 DNA ligase (NEB). The library was purified with AMPure XP beads (Beckmann Coulter), then mixed with the Sequencing Buffer (ONT) and the Loading Bead (ONT), and loaded on a MinION R9.4.1 flow cell. Reads were basecalled using Guppy 3.1.5.

For the Hi-C sequencing, the library was prepared using the Dovetail Omni-C kit (Dovetail Genomics, Scotts Valley, CA, USA). A *P. calceolata* RCC100 culture (60 mL corresponding approximately to $6 \times 10^7$ cells) was first centrifuged at $5000 \times g$ for 10 min. The pellet was processed as mammalian cells, following the Mammalian Cell Protocol for Sample Preparation (version 1.4) without using DSG cross-linking reagent. Briefly, the chromatin was fixed with formaldehyde, randomly digested with DNase I and then extracted. Chromatin ends were repaired and ligated to a biotinylated bridge adapter, followed by proximity ligation of adapter-containing ends. After proximity ligation, crosslinks were reversed and DNA was purified. Purified DNA was treated to remove biotin that was not internal to ligated fragments, and a sequencing library was generated using NEBNext Ultra enzymes and Illumina-compatible adapters. Biotin-containing fragments were isolated using streptavidin beads before PCR enrichment of the library. The Dovetail Hi-C library quality was checked as described above and sequenced on an Illumina MiSeq instrument (Illumina, San Diego, CA, USA) in paired mode (2*150 bp), producing 2,832,092 reads. The Hi-C raw reads were aligned against the assembly (-s none option) using Juicer (Juicer version 1.5.6 - Juicer Tools Version 1.9.9)[49]. The contact map representation was generated with R version 4.1.1 using the merged nodups file.

**RNA extraction, library preparation, and sequencing**. When the cell concentration reached 10 million cells/mL in the mid-exponential growth phase, 160 mL of culture were collected by three successive filtrations on 1.2-µm polycarbonate filters of 47 mm to avoid prokaryotic contamination. To preserve cell and RNA integrity, we kept filtration time and pressure below 10 min and 20 mmHg, respectively. Then filters were stored in 15-mL Falcon tubes with 3 mL of Trizol (Invitrogen, Carlsbad, CA, USA), mixed and flash-frozen in liquid nitrogen for further processing. RNA was extracted by incubation at 65 °C for 15 min, followed by chloroform extraction. The aqueous phase was purified using a Purelink RNA Isolation kit (Ambion Invitrogen, Carlsbad, CA, USA) according to the manufacturer's instructions. DNA contamination was removed by digestion using the TURBO DNA-free™ Kit (Ambion Invitrogen) according to the manufacturer's DNase treatment protocol. After two rounds of 30-min incubation at 37 °C, the efficiency of DNase treatment was assessed by PCR. Quantity and quality of extracted RNA were analyzed with RNA-specific fluorimetric quantitation on a Qubit 2.0 Fluorometer using Qubit RNA HS Assay (Invitrogen). The qualities of total RNA were checked by capillary electrophoresis on an Agilent Bioanalyzer, using the RNA 6000 Nano LabChip kit (Agilent Technologies, Santa Clara, CA).

RNA-Seq library preparation was carried out from 1 µg total RNA using the TruSeq Stranded mRNA kit (Illumina, San Diego, CA, USA), allowing mRNA strand orientation. Briefly, poly(A) + RNAs were selected with oligo(dT) beads, chemically fragmented and converted into single-stranded cDNA using random hexamer priming. After second strand synthesis, double-stranded cDNA was 3'-adenylated and ligated to Illumina adapters. Ligation products were PCR-amplified following the manufacturer's recommendations. Finally, the ready-to-sequence Illumina library was quantified by qPCR using the KAPA Library Quantification Kit for Illumina libraries (KapaBiosystems, Wilmington, MA, USA), and evaluated with an Agilent 2100 Bioanalyzer (Agilent Technologies, Santa Clara, CA, USA). The library was sequenced using 101 bp paired-end read chemistry on a HiSeq2000 Illumina sequencer. Low-quality nucleotides ($Q < 20$) from both ends of the reads were discarded. Illumina sequencing adapters and primer sequences were removed and reads shorter than 30 nucleotides after trimming were discarded. These trimming and cleaning steps were achieved using in-house-designed software based on the FastX package (https://www.genoscope.cns.fr/externe/fastxtend/). The last step identifies and discards read pairs mapped to the phage phiX genome, using the SOAP aligner[50] and the Enterobacteria phage PhiX174 reference sequence

(GenBank: NC_001422.1). This processing, described in ref. [51], resulted in high-quality data. Moreover, ribosomal RNA-like reads were excluded using SortMeRNA[52] 2.1 and SILVA databases.

**Long-read-based genome assembly**. Raw nanopore reads were used for genome assembly. The taxonomic assignation was performed using Centrifuge[53] version 1.0.3 to detect potential contamination. Genome size and heterozygosity rate were estimated using Genomescope[54] and Illumina short reads. For the genome assembly, we generated three sets of ONT reads: all the reads, 30× genome coverage with the longest reads and 30× genome coverage of the highest quality reads estimated by the Filtlong tool (https://github.com/rrwick/Filtlong). We applied Filtlong filtering with default parameters using *Pelagomonas* Illumina short reads as a reference (ONT reads covered by Illumina reads have higher scores). We then applied four different assemblers with default settings, Smartdenovo, Redbean, Flye and Ra on these three sets of reads (Supplementary Data 1)[55–58]. After the assembly phase, we selected the best assembly (Flye with all reads) based on the cumulative size and fragmentation. Indeed, the Wtdbg2 and Smartdenovo assembler generated fragmented assemblies with lower N90. Raven and Flye were very close, but only the Flye assembly with all ONT reads contained both the mitochondrial and chloroplastic circular contigs. To display connections that are not present in the contigs file, we used Bandage tool[59]. The selected Flye assembly was polished three times using Racon[60] with ONT reads, and two times with Hapo-G[61] and Illumina reads. Gene completeness of the assembly was estimated using the single-copy orthologous gene analysis from BUSCO v5 with the stramenopile dataset version 10 containing 100 genes[28].

**Repeat masking and GC analyses**. Repetitive regions on the genome were masked using Tandem Repeat Finder tool[62], Dust tool to detect low-complexity regions[63] and RepeatMasker[64] to identify interspersed repeats based on homology search within the Stramenopile clade and other low-complexity sequences. The positions of detected repeats were merged and hard-masked on the genome, amounting to 8% of its length. Ab initio identification of repeat family sequences was performed using RepeatScout[65]. The algorithm first calculates the frequency of all k-mers in the genome, then removes low-complexity regions and tandem repeats. In >80% of the cases, repeat families identified using ab initio approaches do not overlap with repetitive regions identified by homology search. GC content along the genome was calculated with Bedtools nuc version 2.29.2[66] and the coverage over a non-overlapping window of 2 Kb with Mosdepth version 0.2.8[67].

**Transcriptome assembly**. RNA sequencing reads from *P. calceolata* RCC100 were assembled using Velvet 1.2.07 and Oases 0.2.08 with a k-mer size of 63 bp[68,69]. Reads were mapped back to the contigs with BWA-mem[70] and only consistent paired-end reads were kept. Uncovered regions were detected and used to identify chimeric contigs. In addition, open reading frames (ORF) and domains were searched using respectively TransDecoder (http://transdecoder.sourceforge.net) and CDDsearch[71]. Contig extremities without predicted ORFs or functional domains were removed. Lastly, we used the read strand information to orient RNA contigs. We completed the RNA contigs dataset with the two transcriptome assemblies of the RCC100 strain of *P. calceolata* from the Marine Eukaryotes Transcriptomes database (METdb) (http://metdb.sb-roscoff.fr/metdb/)[72].

**Gene prediction**. Nuclear gene prediction was performed using 23,696 Pelago-monadales proteins (mainly *A. anophagefferens*) downloaded from the NCBI website. Proteins were aligned on the genome in a two-step strategy. First, BLAT[73] (version 36 with default parameters) was used to rapidly localize corresponding putative regions of these proteins on the genome. The best match and the matches with a score greater than or equal to 90% of the best match score were retained. Then, the regions with BLAT alignments were masked and we aligned the same set of proteins using BLAST[74], which can identify more divergent matches. Second, alignments were refined using Genewise[75] (version 2.2.0 default parameters, except the -splice model option to detect non-canonical splicing sites), which is more accurate for detecting intron/exon boundaries. Alignments were kept if more than 50% of the length of the protein is aligned on the genome. Additionally, the transcriptome assemblies of *P. calceolata* RCC969, RCC2362, RCC706 and RCC981 included in the METdb were translated into proteins and aligned to the genome using BLAT, a BLAT score > 50 % filter, and alignments refined with Genewise as previously described.

We selected alignments from the newly generated transcriptome assembly and the two assemblies available in METdb belonging to the *P. calceolata* RCC100 strain to build a training set for the AUGUSTUS ab initio gene predictor[76]. Only gene models with complete coding DNA sequences were retained for training and 1000 genes were set aside for testing AUGUSTUS accuracy. Initial training produced exon and intron parameters for *P. calceolata* species. Parameters were optimized using successive steps of training and testing. We calculated gene prediction accuracy by running AUGUSTUS on the test set. At the exon level, AUGUSTUS performed well in terms of sensitivity (0.619) and specificity (0.669). We thus run AUGUSTUS on the masked genome based on trained parameters.

The ab initio prediction and all the transcriptomic and protein alignments were combined using Gmove, an easy-to-use predictor with no need for a pre-calibration

step[77]. Briefly, putative exons and introns, extracted from predictions and alignments, were used to build a graph, where nodes and edges represent exons and introns respectively. From this graph, Gmove extracts all paths and searches open reading frames (ORFs) consistent with the protein evidence. We trimmed untranslated transcribed regions that overlapped the coding part of a neighbor gene and renamed the genes following the standard nomenclature. Mono-exonic genes models encoding proteins of less than 200 amino acids without significant protein match (1006 genes) were excluded. Chloroplast and mitochondrial genes (contig 7 and 8) were predicted using previously published annotations for *P.calceolata*[23,78]. Following this pipeline, we predicted 16,667 genes with 0.45 intron per gene on average.

**Functional annotation**. Predicted gene models of *P. calceolata* nuclear genome (contig 1 to contig 6) were annotated for protein function using InterProScan v5.41-78.0[79]. A protein alignment against the NR database (01-12-2021 version) was performed with diamond v0.9.24[80]. The best protein match with a functional annotation and an *e*-value < 10$^{-5}$ was retained. KEGG Orthologues (KO) were identified with the HMM search tool KofamKoala v1.3.0 and KO annotations with an *e*-value < 10$^{-5}$ and a score above the HMM threshold were retained[81]. Finally, Gene Ontology (GO) terms and Enzyme commission (EC) numbers were recovered from the Interproscan and KO analysis respectively. Previously published chloroplast and mitochondrial gene names and functions were reported on the corresponding genes[23,78]. All functional gene annotations of *P. calceolata* are available in Supplementary Data 2.

In order to compare the functional annotation of *P. calceolata* with other small free-living photosynthetic eukaryotes, we applied the same analysis to the predicted proteins available for the following species: *Aureococcus anophagefferens*, *Thalassiosira pseudonana*, *Phaeodactylum tricornutum*, *Nannochloropsis oceanica*, *Bathycoccus prasinos*, *Micromonas pusilla*, *Ostreococcus lucimarinus*, and *Emiliania huxleyi* (references are indicated in Table 1).

We defined a list of 23 meiosis-specific genes using three previous studies[31–33]. KO annotations and Interproscan domains were used to recover these genes in the *P. calceolata* genome. Transcriptomic reads of *P. calceolata* were mapped back to the predicted genes with BWA-MEM v 2.2.1. Reads aligned over more than 80% of their length were retained to estimate the relative expression levels of meiosis genes.

Genes containing the NIT-sensing domain were identified based on Interproscan annotations. Eukaryotic homologs of the three *P. calceolata* NIT-sensing genes were retrieved with a BLASTP (*e*-value < 10$^{-5}$, coverage > 100 aa) against 27.7 million proteins from NR, the METdb[72] transcriptome database, eukaryotic algal proteomes from the JGI database, *Tara* Oceans single-cell amplified genomes and metagenome assembled genomes (SMAGs)[82]. The 60 retrieved proteins and the 3 *P. calceolata* NIT-domain-containing proteins were then aligned with Mafft v7.0 (https://mafft.cbrc.jp/alignment/server/) and a Maximum Likelihood phylogenetic tree (Jones–Taylor–Thornton substitution model and 100 bootstraps) was made with MEGAX software. Transmembrane regions in NIT-sensing domain-containing proteins were identified with TMHMM v 2.055.

**Estimation of *P. calceolata* relative abundance in environmental metagenomic reads**. We used metagenomic datasets of *Tara* Oceans and *Tara* Polar Circle expeditions to detect and estimate the relative abundance of the *P. calceolata* in the oceans. Datasets from water samples collected on the photic zone: surface (SUR) and deep-chlorophyll maximum (DCM) were analyzed. Size-fractionated water samples containing pico-nano algae (organisms < 5 μm in cell diameter) were selected: 0.2–3 μm (100 samples), 0.8–2000 μm (119 samples) and 0.8–5 μm (148 samples)[51]. Metagenomic reads were aligned on the *P. calceolata* genome assembled in this study with BWA-mem version 0.7.15 with default parameters[83]. Alignments with at least 90% of identity over 80% of read length were retained for further analysis. In the case of several possible best matches, a random one was picked. In order to remove putative PCR duplicates, multiple read pairs aligned at the same position on the *P. calceolata* genome were removed with samtools rmdup version 1.10.2[70]. For the metagenomic abundance, we divided the total number of reads aligned on the *P. calceolata* genome assembled in this study by the total number of sequenced reads for each sample. For the metabarcoding abundance, we used the 18SV9 rRNA OTU table published in 2019 and available here https://zenodo.org/record/3768510#.YEX2S9zjJaQ[34]. Bacterial and archaea OTUs were removed for this analysis.

**P. calceolata relative abundance models**. Two *P. calceolata* relative abundance models were performed based on the in situ environmental conditions measured at the time of sampling or using the WOA18 datasets. The environmental parameters measured during the expedition are available in the Pangaea database (https://www.pangaea.de/) and are described in ref. [84]. Iron concentrations are annual means derived from PISCES2 model[85] and described in ref. [13]. Ammonium concentrations at the date and location of sampling are derived from the MITgcm Darwin model and available in the Pangaea database[86]. Environmental parameters for each sample are available in Supplementary Data 10. We consider oceanic samples as "low-iron" if they contain less than 0.2 nM of iron, "low-nitrate" if they

contain less than 2 μM of nitrate and "low-ammonium" if they contain less than 25 nM of ammonium. These thresholds were defined with the distribution of nutrient concentrations in the dataset and previous studies[13,14]. Pearson's correlations between the relative abundance of *P. calceolata* and all environmental parameters were calculated with the *cor* function the R package FactoMineR version 2.4 and the GGally package version 2.1.0. Principal component analysis (PCA) was performed with 9 parameters presenting significant Pearson's correlations. We used a Generalized Additive Model (GAM) for its ability to fit non-linear and non-monotonic functions and for its low sensitivity to extreme values to model the relative abundance of *P. calceolata* as a function of iron concentration, temperature and PAR light[87]. This function is implemented in the mgcv R package version 1.8–33.

The mean of several climatologies of the Earth System Models under the RCP8.5 scenario (ESM8.5) was used to define the environmental conditions at the end of the century following the method of ref. [35]. *P. calceolata* relative abundance models based on the WOA18 at locations, depths, and months of *Tara* Oceans samples or ESM8.5 were obtained using four machine-learning techniques as described in ref. [35]. with some differences. The four tools were trained in regression mode, we used the neuralnet R package[88] with the following parameters: hidden = 1, decay = 0.1, mxit = 1e$^6$ and size = 5, 6 splines were selected for the GAM. We performed 100-fold random cross-validation for each model and evaluated their performance using the Pearson correlation coefficient and rmse. We used the ensemble model approaches[35] for final global-scale models of the relative abundance of *P. calceolata* (i.e., the mean projections of the validated machine-learning techniques). Figures 2a, 3c, and Supplementary Fig. S7 were generated using the world map of R package "maps", "mapdata" and "ggmap". World maps data are imported from the public domain Natural Earth project.

**Environmental metatranscriptomic reads mapping and filtering**. Metatranscriptomic reads from *Tara* Oceans datasets were aligned on predicted coding sequences of the *P. calceolata* genome with BWA-mem 2.2.1 using default parameters. We selected reads aligned with more than 95% of identity over 80% of the read length. We kept nuclear genes covered by at least ten reads in a minimum of ten samples and removed those detected in more than 90% of samples which probably aggregate metagenomic reads from other organisms (cross-mapping). To obtain robust relative expression values, we removed samples where less than 25% of *P. calceolata* genes were detected. We finally got the expression of 15,679 genes of *P. calceolata* across 167 samples. In all gene expression analyses, we normalized the gene expression levels in transcripts per kilobase million (TPM). Gene expression levels of *P. calceolata* in all *Tara* Oceans samples are available here: https://doi.org/10.5281/zenodo.6983365.

**Statistical analysis and reproducibility**. Sample sizes are indicated in the methods, in the figure captions or in the main text. Nonparametric Wilcoxon signed-rank tests were applied with the two-sided alternative hypothesis and not paired. Two sides Fisher statistical tests were applied for gene enrichments analysis. All statistical tests in the manuscript were generated with R version 4.0.3 and *P* values < 0.01 are considered significant.

**Reporting summary**. Further information on research design is available in the Nature Research Reporting Summary linked to this article.

## Data availability

*Pelagomonas calceolata* genomic and transcriptomic reads, genome assembly, and gene prediction are available at the ENA (EMBL-EBI) website under the accession number PRJEB47931. *P. calceolata* transcriptomes are available under the accession number PRJEB34158, runs ERR3497221 and ERR3497222. *Tara* Oceans and *Tara* Polar Circle metagenomic sequences are archived at the ENA under the following accession numbers: PRJEB9740, PRJEB9691, PRJEB4352, and PRJEB1787. All other data are available from the corresponding author upon reasonable request.

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

## Acknowledgements

We thank the commitment of the following people who made this work possible: the Genoscope/CEA, the CNRS (in particular the Federation de Recherche R2022/Tara Oceans GO-SEE), Marie-José Garet-Delmas from the Roscoff Culture Collection for growing the RCC100 strain, Claude Scarpelli for support in high-performance computing and Linda Sperling for language editing. Computations were performed using the cobalt HPC machine. We acknowledge the financial support of FRANCE GENOMIQUE (ANR-10-INBS-09–08) and Oceanomics (ANR-11-BTBR-0008). We also thank the *Tara* Expedition Foundation and their partners for the organization of marine scientific expeditions (http://oceans.taraexpeditions.org). This article is contribution number 139 of *Tara* Oceans.

## Author contributions

S.R., C.B., and M.G. performed *P. calceolata* cultures and DNA/RNA extractions. A.A., E.P., and C.C. coordinated DNA/RNA sequencing. B.I, B.N., and J.M.A. did the assembly and annotation of the genome. M.C., S.M., Q.C., and J.M.A. carried out genomic analysis. N.G., E.F., and Q.C. analyzed environmental datasets. P.F. and O.J. worked on *P. calceolata* models. Q.C. and N.G. wrote the paper with a strong support of J.M.A. and P.W. C.B, L.B., and K.L. did the Hi-C library preparation, sequencing and analysis, respectively. All authors contributed to the manuscript preparation and approved the final version of the paper.

## Competing interests

The authors declare no competing interests.
