## [Peer Review File · Communications Biology]

Reviewers' comments:

Reviewer #1 (Remarks to the Author):

It is of interest to sequence another pelagophyte species (after *Aureococcus anophagefferens*). Overall the genome sequencing, assembly, and annotation appear to have been reasonably well done. However, the ecological inference of the genomic data is weak, particularly because the TARA Oceans metatranscriptomic data were not duly utilized, although the authors did use the metagenomic data to profile species distribution and relative abundance. Besides, there is apparently an error in depicting nitrogen uptake and metabolic pathways. Finally, there lacks independent analysis to verify that the key functional genes highlighted in the manuscript are from *P. calceolate* rather than contaminating bacteria, which requires phylogenetic inferences.

Specific comments are listed below.

TARA Oceans has huge metatranscriptome dataset, which should be helpful to find if the *Pelagomonas calceolate* genes reported in this manuscript as being adaptive to habitats (e.g. iron deficient) are expressed in the way as expected. This would nicely complement what the authors have done with the abundance of the species.

How relative abundance was calculated in the global ocean should be explained in Methods. How was genome number counted from the metagenomic data without the ability of getting whole genomes assembled?

Line 368, "Among 20 genes specifically involved in meiosis": based on what source of information? Provide reference.

Figure 2: the authors seem to assume that ammonium only goes to urea cycle, which occurs in mitochondria, and does not enter the chloroplast to be assimilated by GS. This is incorrect, best to my knowledge.

P. calceolate is the third most abundant eukaryote of the 0.8-5 micron size fraction in TARA samples, what are the first and the second? Provide the information.

Line 452: how can you be sure it was an estimate by rDNA and not overestimate by genome-based estimation?

Discussion

Explain if the low-GC regions are located in positions anticipated for centromeres.

Related to an earlier comment, the expression profile of iron stress-coping genes (ISIP2, ISIP3, ferroportins) should be examined from the TARA Oceans dataset in order to link the relative abundance of *P. calceolate* to iron poor condition.

Similar to the comment above, expression profiles should be examined for nitrate/nitrate versus ammonium uptake genes in order to draw a tentative conclusion that *P. calceolate* might dominantly utilize nitrate/nitrite instead of ammonium.

Reviewer #2 (Remarks to the Author):

Despite the ecological relevance of photosynthetic picoeukaryotes, reference genomic data for widespread species is still scarce. Pelagophytes are known to be major components of PPE assemblages and are widely distributed. Molecular surveys based on ribosomal genes revealed that most environmental sequences retrieved for this group are related to *Pelagomonas calceolata*. In this manuscript the authors sequenced the genome and the transcriptome of a cultured strain of *P. calceolata* to evaluate its genome capability focusing on genes related to sex, and nitrogen and iron metabolisms. They also evaluated its relative abundance in globally distributed samples from

the Tara-Oceans expedition mapping metagenomic reads to the assembled genome and compared it with the abundances obtained based on metabarcoding data (18SV9 rRNA gene) from the same survey. This is a relevant manuscript contributing to a better understanding of PPEs functional capabilities, and presenting *P. calceolata* as a model organisms for futures studies on the ecological role of PPEs.

The manuscript is very well written, the methods have been explained in detail and are appropriate for addressing the questions presented in the study. Results are presented concisely, pointing to the main conclusions for each section. The authors did a great effort to compile all the information and presented relevant analysis as for instance modelled abundances for *P. calceolata*. Nevertheless, some clarifications should be done regarding 1) the RNA seq results. It was not clear for me which results in the manuscript are the outcome of the transcriptome assembly and which ones of the long-read + short reads genome assembly; 2) the relative abundance estimates based on metabarcoding and metagenomics. Also, I missed a direct comparison and discussion about the gene content (nitrogen and iron metabolisms) between *Pelagomonas* and *Aureococcus*. Finally, I think that one of the main conclusions of the paper as stated in the title, should be supported by a definition of what is considered iron-poor waters (concentration ranges and which Tara stations would fit into that classification) but this was not addressed in the text.

Here my specific comments for the authors:

- Introduction:

L44. Please consider to extend this size to $\leq 3 \mu\text{m}$, as many important PPE groups are mostly represented by 2-3 μm forms (Vaulot et al., 2008; Jardillier et al., 2010; Grob et al., 2011).

- Methods:

Please add references for Centrifuge 1.0.3, BLAT, BLAST, Genewise, AUGUSTUS... also, revise abbreviations along the text (ie., ONT, NIT-sensing, NR etc...) to include the complete name when they appear for the first time (I think methods section in COMMSBIO goes at the end of the manuscript).

L176. Highest quality-scored?

L216. Do you mean BLAT or BLAST?

L219. In line 202 you use RCC100 for the name of the strain. I think both names are equivalent but keep just one along the text.

L235. Do you mean introns?

L261. "...single -cell amplified genomes and metagenome assembled genomes...", replace "and" for a comma?

L288. Add in the text: "...are described in Pesant et al.," not only the number of the reference. Same issue in line 289.

L313. Add the strain name.

L326. Could you explain better what is observed on figure 2A?

L333. Add "s" to method.

- Results:

L382. Is this result section the outcome of the transcriptome sequencing? Please define more clearly which sequencing approaches (DNA seq or RNA seq) were used for the different sections of the manuscript (both in the methods sections and in the results sections). It was not clear for me for instance which dataset did you use to perform the functional analysis (methods section).

Table S7. Include the number of genes for the other 8 species. How different is the gene content of *Aureococcus*? Is any of the algae listed in table 1 also widespread in oceanic regions? I am wondering whether the gene repertoire (for nitrate and iron metabolisms) of any of these algae is similar to *P. calceolata* even though they are more coastal related species. Please include a comment about this on the discussion section L-551.

L441 Clarify why the relative abundance of *P. calceolata* in Tara metabarcoding samples for the 0.2-3 μm size fraction was not determined.

L444-L446. In Table S8, 0.81% corresponds to surface and 1.3% corresponds to DCM. Please correct accordingly in the text; also the total number of samples (150 in the table 111 in the text). Are you presenting here the most abundant OTUs? If so, specify that in the table heading. Looking and the Tara methods I think this OTU table is based on swarms. I am wondering whether there were other less represented OTUs also assigned as *Pelagomonas*, as well as for the other species. If so, that would change the numbers and the statement "...*P. calceolata* is the third most abundant eukaryote..."(L446). Is the abundance of this specific *Pelagomonas* OTU what you are considering for the 18sV9 data represented on figure S6?

Figure S6. Please, specify the number of samples represented for each size fraction and add the equation for the regression line. Add a supplementary table with the mean relative abundance values (+SD) represented on this figure for surface, DCM in both size fractions and for both metabarcoding and metagenomic datasets.

L457. Add in the text or in a supplementary table the mean (+SD) relative abundance values at surface and DCM in the 3 size fractions, so the reader can have a summary of the abundances obtained with the metagenomic approach.

L458. Please clarify if this 1% is considering all size fractions or just the 0.8-5 μm fraction. Also, clarify that the 6.7% is for a specific sampling station.

Figure 3. Please, add "metagenomic reads" in the figure legend: A) World map of the relative abundance of *P. calceolata* metagenomic reads.

L468. One of the measured parameters was the concentration of 9'butanoyloxyfucoxanthin, a signature pigment for pelagophytes. I am just curious to know whether you checked the correlation between this pigment and the read relative abundances.

L474. In Table 3, please add the same analysis for the 0.2-3 μm size fraction.

L476. I would say, "The higher relative abundance..." instead of "The high...". To state that, it would be helpful to specify which value of iron concentration generally determines iron-poor waters and give average relative abundances below and above this value. Support this with a statistic test (Mann-Whitney test, for example).

-Discussion:

L543, Include number for qPCR estimates

Reviewer #3 (Remarks to the Author):

The manuscript by Guérin et al. describes generation, analysis, and utilization of the genome sequence of *Pelagomonas calceolata*, a widely distributed photosynthetic marine picoplankton. In the first part of the manuscript, the authors present details of the sequencing approach and genomic structure and provide some high-level analysis of genome content. The second part of the manuscript maps publicly available metagenomic sequences to the *P. calceolata* genome and uses this information to estimate current abundance distributions and to predict distributions into the future. In its current state, the manuscript reads like two papers with a tenuous link between the two. For example, it is not clear how the genome structure information (introner element distributions, GC distribution across chromosomes, impact of presence/absence of meiosis genes) from the first half of the paper are related to the second half of the paper that focuses on *P. calceolata* distributions in current and future oceans. The end result is that both halves of the

paper could be more fully analyzed. Below, I highlight areas for improvement.

1) The introduction consists of 5 paragraphs on changing ocean conditions, a single paragraph on *Pelagomonas* and a single paragraph on what the authors focused on in the manuscript. There is essentially no rationale presented for why a chromosome-resolved genome is required for understanding the distribution of *P. calceolata* in global oceans. What does the complete genome provide that is not provided by transcriptome or metatranscriptome data?

2) Methods.

A) The assembly step could benefit from more detail than "we selected the best assembly (Flye with all reads) based on the cumulative size and contiguity." I do not see any indication of contiguity in Table S1. Do the different assemblies rely on default settings? They each result in vastly different numbers of contigs. A brief description of the evidence that the longest contigs are the correct assemblies rather than a merger of different chromosomes. A prediction of 6 chromosomes is and 'outlier' relative to the chromosome complement of other sequenced picoplankton as shown in Table 1. An independent verification of aspects of the assembly should be included, such as some PCR-based assessments in different regions.

B) The authors should present the results of their BUSCO-based analysis to assess genome completeness.

C) The description of the models should be expanded, particularly how model performance was evaluated. There is relatively little information provided for a major component of the manuscript, which is predicting relative abundance in 2099.

3) Lines 312-331. The first sentence of the results section illustrates the limited rationale the authors present for these studies. "To investigate its gene repertoire and its distribution across the oceans we sequenced and assembled the genome of *P. calceolata* using ONT long-reads and Illumina short-reads." The authors then document details of how the sequence was generated.

4) Lines: 330-331. The authors present the predicted number of genes in the *P. calceolata* genome. In Fig. S4, they also present the number of *P. calceolata* proteins homologous to proteins from other stramenopile genomes. The authors should also provide the results of a comparison between genes identified via transcriptome assemblies (from both the MMETSP dataset and their own transcriptomes) and genes identified via their gene prediction algorithms. For example, they should provide a table that includes basic stats such as the number transcriptome-assembled genes detected in the genome assembly, the number of transcriptome-assembled genes not detected in the genome assembly, the number of genome-assembled genes not detected in the transcriptome assemblies. The latter number is important as a way of further justifying the power of a complete genome assembly rather than transcriptomes from a variety of conditions. For example, were the "meiosis-specific" genes detected only in the genome assembly and not the transcriptome assembly?

5) The results on the predicted gene composition of the low-GC regions are interesting.

6) Line 381. The authors hypothesize that *P. calceolata* can undergo meiosis based on the genome complement and genome structure. The authors should discuss whether there is transcriptome support for any of the meiosis genes and if so, under what conditions. For example, are there transcripts associated with any of these genes under exponential growth conditions and if so, what are the implications?

7) Fig. 2. The authors compared the number of nitrogen utilization genes detected in *P. calceolata* vs the average number of these genes in seven other picophytoplankton and used the comparison to identify genes as "overrepresented," "equally represented," or "absent." The comparison to the average is misleading. Instead, the authors should compare to the range of values found in the other organisms. Only two genes highlighted in their nitrogen analysis are outside the range of values found in other organisms: glutamate synthase is present in 4 copies, which is outside the range of 1-3 in other organisms and nitrate reductase is present in 1 copy, which is outside the range of 2-5 in other organisms. It is a stretch to conclude from a comparison to the mean that "Among them, 8 genes encode nitrate/nitrite or formate/nitrite transporters, which is on average higher than in other algae. In contrast, only 5 genes encode ammonium transporters which is low compared to other species suggesting that nitrite and/or nitrate is the main external source of inorganic nitrogen for *P. calceolata*." At a minimum, the authors should discuss the biogeography implications if these organisms preferentially use nitrate/nitrite relative to ammonium.

8) Lines 412 -414. The authors should clarify the conclusion that "*P. calceolata* is not particularly adapted to recycle nitrogen from organic molecules but could be capable of incorporating inorganic nitrogen compounds even in N-poor environments." Do the authors mean to imply that there is

sufficient nitrate/nitrite in N-poor environments to support growth?

9) Line 429-430. The authors state "the absence of these genes [FTR1, ISIP1, ferritin] suggests that iron uptake and storage is not a major asset of *P. calceolata* compared to the other photosynthetic protists." This speculative statement should be backed up by physiological data. There are many picophytoplankton without ferritin that nonetheless can grow under a wide range of iron concentrations.

10) The comparison of metabarcoding vs genome mapping to estimate relative abundance is interesting. Are there any differences if the mapping is against the available transcriptomes?

11) Lines 465 -470. It is surprising to me that PAR is not related to *P. calceolata* abundance, especially given that the distribution data that suggests a greater relative abundance at the DCM, particularly in the Indian Ocean. In addition, the authors infer that nitrate concentrations are important for growth of *P. calceolata*. These concentrations are expected to be higher at the DCM.

12) Lines 477-478. The authors should also clarify their conclusion based on model results that "the high relative abundance of *P. calceolata* in iron-poor waters suggests that this species is particularly capable of acclimation to this environmental condition." Lines 429-430 (see comment #8) suggest that "iron uptake is not a major asset of *P. calceolata*..."

13) Line 593-595. The authors should temper their conclusion "the chromosome- scale genome sequence, mostly telomere-to-telomere, generated in this study is an essential starting point for its detection in environmental datasets." The authors have not convinced me that the telomere-to-telomere genome sequence is required to detect organisms in environmental datasets.

Response to referees

*Author answers are in blue. Citations from the manuscript file are in italic.

Reviewer #1

It is of interest to sequence another pelagophyte species (after *Aureococcus anophagefferens*). Overall the genome sequencing, assembly, and annotation appear to have been reasonably well done. However, the ecological inference of the genomic data is weak, particularly because the TARA Oceans metatranscriptomic data were not duly utilized, although the authors did use the metagenomic data to profile species distribution and relative abundance. Besides, there is apparently an error in depicting nitrogen uptake and metabolic pathways. Finally, there lacks independent analysis to verify that the key functional genes highlighted in the manuscript are from *P. calceolate* rather than contaminating bacteria, which requires phylogenetic inferences.

Specific comments are listed below.

TARA Oceans has huge metatranscriptome dataset, which should be helpful to find if the *Pelagomonas calceolate* genes reported in this manuscript as being adaptive to habitats (e.g. iron deficient) are expressed in the way as expected. This would nicely complement what the authors have done with the abundance of the species.

Although the gene expression patterns of *P. calceolata* was not the primary topic of this study, we agree that transcriptomic datasets are interesting for genes potentially involved in the adaptation to the environment. Therefore, we analysed the gene expression of genes involved in iron and nitrogen metabolisms across the oceans. Among these genes, we identified those with a relative expression variable according to the environment (nitrate or iron concentration). The results bring out important conclusions on the acclimation capacities of *P. calceolata* and support most of our hypotheses based on the genome. These new data are presented in the iron and nitrate paragraphs, we added a main figure (Figure 4) and a new panel to Figure 5 as well as 3 supplementary figures (Figures S8-S10).

How relative abundance was calculated in the global ocean should be explained in Methods. How was genome number counted from the metagenomic data without the ability of getting whole genomes assembled?

The genome-based relative abundance of *P. calceolata* in a sample is obtained with the number of metagenomic reads aligned on the *P. calceolata* genome divided by the total number of reads sequenced for this sample. The percentage of DNA belonging to a species in a sample is a proxy for its relative abundance. This abundance estimation is dependent on the genome size (large genomes are overrepresented). Lines 154-157 in the result section and 560-561 in the Mat&Met were modified to precise this point.

Line 368, "Among 20 genes specifically involved in meiosis": based on what source of information? Provide reference

The three articles used to define this list of meiosis-specific genes are indicated in the Mat&Met (lines 535-539), Chi, et al 2014; Ramesh et al ; Schurko et al 2008. To improve clarity, we added these references in the result section and modified the sentence line 137-138. In addition, we corrected the number of genes indicated in the text based on Table S7: *Among 23 meiosis specific genes characterised in other species³⁸⁻⁴⁰, 18 homologs are present in the P. calceolata genome (Table S7).*

Figure 2: the authors seem to assume that ammonium only goes to urea cycle, which occurs in
mitochondria, and does not enter the chloroplast to be assimilated by GS. This is incorrect, best to
my knowledge.

We thank the referee for spotting this mistake. We added the missing arrow in this figure (now
Figure 5).

*P. calceolate* is the third most abundant eukaryote of the 0.8-5 micron size fraction in TARA samples,
what are the first and the second? Provide the information.

This information is provided Table S8. We added the name of these 2 lineages in the text lines 151-
153: *According to this method of abundance estimation, P. calceolata is the third most abundant*
*eukaryote OTU of the 0.8-5 μm size-fraction after two Dinophyceae OTUs affiliated to Ankistrodinium*
*and to an unknown Gymnodiniaceae.*

Line 452: how can you be sure it was an estimate by rDNA and not overestimate by genome-based
estimation?

The rRNA-based abundance estimation is dependent on the number of rRNA gene copies in each
genome. The number 18S rRNA genes in *P. calceolata* is low (only 2 copies) compared to most of other
eukaryotes and therefore the abundance of this species is probably underestimated with the
metabarcoding. Genome-based estimation is dependent on genome sizes: organisms with larger
genomes are overrepresented. *Pelagomonas* genome is relatively small compared to other eukaryotes
but is large compared to prokaryotes that are also present in the 0.8 - 5 μm size-fraction. It is then
difficult to say if this method will overestimates or underestimates the abundance. We removed this
sentence in the result and changed the paragraph in the discussion lines 295-299: *In addition, we have*
*shown that the metabarcoding approach probably underestimates the relative abundance of P.*
*calceolata compared to the metagenomic analysis owing to the low copy number of rRNAs in this*
*organism. However, we cannot exclude that the large genome size of P. calceolata compared to*
*bacterial genomes present in this size-fraction overestimate its relative abundance in metagenomic*
*datasets.*

Discussion

Explain if the low-GC regions are located in positions anticipated for centromeres.

We have no *a priori* on the position of centromeres in the *P. calceolata* genome. We have two
arguments to say that these low-GC regions contain the centromeres: 1) we have a single low-GC
region per chromosome. 2) The Hi-C result added in the revised manuscript shows physical proximity
of low-GC regions across chromosomes (Figure S2, Supplementary Note 2). Indeed, centromeres co-
localised in the nucleus architecture (Mizuguchi T., et al 2014) and the Hi-C method is able to reveal
this proximity (Varoquaux et al 2015, Muller et al 2019).

Related to an earlier comment, the expression profile of iron stress-coping genes (ISIP2, ISIP3,
ferroportins) should be examined from the TARA Oceans dataset in order to link the relative
abundance of *P. calceolata* to iron poor condition.

We analyzed the expression profiles of iron stress genes and reported the expression profiles in low
versus high iron conditions in the new Figure 4 and in Supplementary Figure S8. Most of ISIP genes are
overexpressed in low-iron conditions. Ferroportins are not differentially expressed. Results are
described in the iron paragraph lines 213-218 and discussed lines 314-323.

Similar to the comment above, expression profiles should be examined for nitrate/nitrate versus
ammonium uptake genes in order to draw a tentative conclusion that *P. calceolata* might dominantly
utilize nitrate/nitrite instead of ammonium.

We analyzed the expression profiles of nitrate genes and reported the expression profiles in low versus
high nitrate conditions in the new panels of Figure 5 and in Figures S9, S10. One formate/nitrite and 2
nitrate/nitrite transporter genes are significantly overexpressed in low-nitrate conditions. One
ammonium transporter is overexpressed in low-ammonium environments. Based on these new results
we can not affirm that the preferred source of inorganic nitrogen is the ammonium, therefore we
removed this conclusion. Results are described in the nitrogen paragraph lines 246-249.

**Reviewer #2 (Remarks to the Author):**

Despite the ecological relevance of photosynthetic picoeukaryotes, reference genomic data for
widespread species is still scarce. Pelagophytes are known to be major components of PPE
assemblages and are widely distributed. Molecular surveys based on ribosomal genes revealed that
most environmental sequences retrieved for this group are related to *Pelagomonas calceolata*. In this
manuscript the authors sequenced the genome and the transcriptome of a cultured strain of *P.*
*calceolata* to evaluate its genome capability focusing on genes related to sex, and nitrogen and iron
metabolisms. They also evaluated its relative abundance in globally distributed samples from the
Tara-Oceans expedition mapping metagenomic reads to the assembled genome and compared it
with the abundances obtained based on metabarcoding data (18SV9 rRNA gene) from the same
survey. This is a relevant manuscript contributing to a better understanding of PPEs functional
capabilities, and presenting *P. calceolata* as a model organisms for futures studies on the ecological
role of PPEs.

The manuscript is very well written, the methods have been explained in detail and are appropriate
for addressing the questions presented in the study. Results are presented concisely, pointing to the
main conclusions for each section. The authors did a great effort to compile all the information and
presented relevant analysis as for instance modelled abundances for *P. calceolata*. Nevertheless,
some clarifications should be done regarding

1) the RNA seq results. It was not clear for me which results in the manuscript are the outcome of the
transcriptome assembly and which ones of the long-read + short reads genome assembly;

In the first version of the manuscript, all results were based on the genome: RNAseq reads were only
used to determine the gene position on the genome sequence. In the revised version of the
manuscript, environmental metatranscriptomic data were analysed.

2) the relative abundance estimates based on metabarcoding and metagenomics.

The relative abundance estimates are explained in the result section lines 147-169 and in the method
lines 549-564. See below for more details.

Also, I missed a direct comparison and discussion about the gene content (nitrogen and iron
metabolisms) between *Pelagomonas* and *Aureococcus*.

We completed Table S9 to indicate the gene content related to iron and nitrate metabolism for the 8
species separately. We added several sentences in the results and in the discussion to compare directly
*Pelagomonas* and *Aureococcus*. The main differences are the absence of ISIP3 gene and a lower
number of Zinc-Iron permeases in *Aureococcus* (lines 223-224 and 312-314), the high number of

Flavodoxin/ferredoxin genes in *P. calceolata* (lines 227-229) and the reduced number of ammonium
transporter in *P. calceolata* (lines 331-332).

Finally, I think that one of the main conclusions of the paper as stated in the title, should be
supported by a definition of what is considered iron-poor waters (concentration ranges and which
Tara stations would fit into that classification) but this was not addressed in the text.

We defined as low-iron environments concentrations below 0.2 nmol/l. We indicated this threshold
in the Mat&Met line 598-602 and in the Result section line 211-212.

Here my specific comments for the authors:

- Introduction:

L44. Please consider to extend this size to $\leq 3 \mu\text{m}$, as many important PPE groups are mostly
represented by 2-3 μm forms (Vaulot et al., 2008; Jardillier et al., 2010; Grob et al., 2011).

We agree with the referee and removed the “picoplankton” terminology (defined by cells between
0.2 and 2 μm , Raven et al 1998), line 43.

- Methods:

Please add references for Centrifuge 1.0.3, BLAT, BLAST, Genewise, AUGUSTUS... also, revise
abbreviations along the text (ie., ONT, NIT-sensing, NR etc...) to include the complete name when
they appear for the first time (I think methods section in COMMSBIO goes at the end of the
manuscript).

Missing references were added. We moved the method section after the discussion and the
abbreviations were added at their first occurrence when missing.

L176. Highest quality-scored?

The read score is indeed based on the Phred quality. We changed the sentence line 453.

L216. Do you mean BLAT or BLAST?

There is no mistake here. BLAT is computationally faster and its sensitivity is similar to BLAST when the
query (*P. calceolata* transcriptome) and the target (*P. calceolata* genome) belong to the same species.

L219. In line 202 you use RCC100 for the name of the strain. I think both names are equivalent but
keep just one along the text.

The two strain names are indeed synonymous. We kept RCC100.

L235. Do you mean introns?

The sentence is correct: in average, there is 1.45 exon per gene in the genome, so 0.45 introns per
gene. To avoid any confusion, we modified the sentence to indicate the number of introns instead of
the number of exons (line 114 and 516).

L261. “...single -cell amplified genomes and metagenome assembled genomes...”, replace “and” for a
comma?

L288. Add in the text: “...are described in Pesant et al.,” not only the number of the reference. Same
issue in line 289.

L313. Add the strain name.

The 3 sentences were corrected.

L326. Could you explain better what is observed on figure 2A?

Because there is no Figure 2A we suppose that the reviewer means Figure S2A (Figure S1b in the
revised version). This figure is the representation of nanopore read assembly with Flye. We modified
the figure legend to provide more details: *Graphical representation of the Flye assembly graph
generated with Bandage tool (Wick et al. 2015). Each coloured box represents a sequence (edge) of the
assembly. Edges connected with one or several black lines indicate unresolved repeats at the extremity
of the contig (e.g edge_2 is connected to edge_3 and/or edge_5. Based on the vertical coverage, Flye
chose to duplicate edge_2 to form contig 3 and contig 6).*

L333. Add "s" to method.

Mistake corrected.

- Results:

L382. Is this result section the outcome of the transcriptome sequencing? Please define more clearly
which sequencing approaches (DNA seq or RNA seq) were used for the different sections of the
manuscript (both in the methods sections and in the results sections). It was not clear for me for
instance which dataset did you use to perform the functional analysis (methods section).

We modified these paragraphs to be clear on the data used to performed the analysis :

line 210-213: *Since P. calceolata seems to thrive in iron-poor waters, we identified P. calceolata genes
coding for iron uptake and storage then, compared their expression levels in low (<0.2 nmol/l) versus
high (>0.2 nmol/l) iron conditions using Tara Oceans metatranscriptomes.*

line 241-243: *Because P. calceolata could be an important player in the nitrogen (N) cycle in oceanic
ecosystems²⁴, we explored its gene content and analysed the expression levels of genes involved in
nitrogen metabolism in the environment using Tara Oceans metatranscriptomes (Figure 5 and Table
S9).*

Table S7. Include the number of genes for the other 8 species. How different is the gene content of
Aureococcus? Is any of the algae listed in table 1 also widespread in oceanic regions? I am wondering
whether the gene repertoire (for nitrate and iron metabolisms) of any of these algae is similar to *P.*
*calceolata* even though they are more coastal related species. Please include a comment about this
on the discussion section L-551.

We modified this table (now Table S9) to include the number of genes in each of the 8 other species.
The 3 chlorophyte species (*Bathycoccus prasinos*, *Micromonas pusilla* and *Ostreococcus lucimarinus*)
and the haptophyte *Emiliania huxleyi* are also widespread in the open oceans. The other Stramenopiles
(*Nannochloropsis*, *Phaeodactylum*, *Thalassiosira* and *Aureococcus*) dominate in coastal ecosystems.
The gene repertoire most similar to *Pelagomonas* is that of *Aureococcus* which is not surprising given
their phylogenetic proximity. However, we identified important differences (ISIP, iron permeases,
Flavodoxin/Feredoxins) potentially explaining the different ecological niches for these two species (see
comments above). We added a sentence in the discussion line 318-320: *Compared to P. calceolata, the
low abundance of Aureococcus in open oceans where iron is limited could be explained by the absence
of genes involved in iron uptake and storage, including iron permeases and ISIP3 genes.*

L441 Clarify why the relative abundance of *P. calceolata* in Tara metabarcoding samples for the 0.2-3
210 μm size fraction was not determined.

18S rRNAs were not sequenced for the 0.2-3 μm size-fraction in *Tara* Oceans samples because it
contains mostly prokaryotes. Therefore, eukaryote abundances are not available for this size-fraction.

L444-L446. In Table S8, 0.81% corresponds to surface and 1.3% corresponds to DCM. Please correct
accordingly in the text; also the total number of samples (150 in the table 111 in the text).

We revised Table S8. A few samples corresponding to RNA instead of DNA abundances were removed
from the OTU table before the calculation of the relative abundance. Relative abundances of OTU were
sorted according to all 0.8 - 5 μm size-fraction samples (whatever the sampling depth). The result is 104
surface and 61 DCM samples with a relative abundance of 0.8% and 1.23% respectively. The
manuscript was modified accordingly lines 149-153.

Are you presenting here the most abundant OTUs? If so, specify that in the table heading.

Table S8 heading was modified according to the reviewer remark.

Looking and the Tara methods I think this OTU table is based on swarms. I am wondering whether
there were other less represented OTUs also assigned as *Pelagomonas*, as well as for the other
species. If so, that would change the numbers and the statement "...*P. calceolata* is the third most
abundant eukaryote..."(L446).

The OTU table is indeed based on swarm. There is a total of 118 OTUs affiliated to *Pelagomonas*. The
2nd most abundant OTU affiliated to *P. calceolata* is 1000 times less abundant than the main OTU
(0.00067% vs 0.92%) in average of all 0.8 – 5 μm size-fraction. We do not know if these minor OTUs
are PCR mistakes, swarm artefacts, *Pelagomonas* sub-populations or other *Pelagomonas* species
therefore, we cannot sum up the abundance of these OTUs. For the other OTUs, the taxonomic
affiliation is often at the genus or family level so we cannot evaluate the species abundance. We
modified the text line 148-150 to precise this point : *The most abundant P. calceolata Operational*
*Taxonomic Unit (OTU) in the 0.8-5 μm size-fraction is on average 0.80% for the 104 surface samples*
*and 1.23% for the 61 deep-chlorophyll maximum (DCM) samples (Table S8).*

Is the abundance of this specific *Pelagomonas* OTU what you are considering for the 18sV9 data
represented on figure S6?

In this figure (now Figure S5), only the most abundant OTU is considered for the reasons explained
above.

Figure S6. Please, specify the number of samples represented for each size fraction and add the
equation for the regression line. Add a supplementary table with the mean relative abundance values
(+SD) represented on this figure for surface, DCM in both size fractions and for both metabarcoding
and metagenomic datasets.

The number of samples and the equations of the regression lines were added to the figure. The number
of samples is lower than in Table S9 because only samples with both metabarcoding and metagenomic
sequencing data are presented.

L457. Add in the text or in a supplementary table the mean (+SD) relative abundance values at
surface and DCM in the 3 size fractions, so the reader can have a summary of the abundances
obtained with the metagenomic approach.

We added the mean and standard deviation in the text line 155-158: *The percentage of sequenced*
 *reads aligned on the genome is 1.39% (sd = 1.5) in surface samples and 2.67% (sd = 1.6) in DCM samples*
 *for the 0.8-5 µm size fraction. In the 0.8-2000 µm size fraction, P. calceolata represents 1.01% (sd = 1.2)*
 *of all reads in surface samples and 1.56% (sd = 1.3) in DCM samples.*

L458. Please clarify if this 1% is considering all size fractions or just the 0.8-5 µm fraction. Also, clarify
 that the 6.7% is for a specific sampling station.

We replaced this 1% by the average as suggested above. We added the name of the station line 158-
 160 : *A maximal relative abundance of 6.7% in the 0.8 - 5 µm size-fraction was observed in the North*
 *Indian Ocean (station TARA_38) at the DCM (Figure 2a).*

Figure 3. Please, add “metagenomic reads” in the figure legend: A) World map of the relative
 abundance of *P. calceolata* metagenomic reads.

Legend modified according to reviewer suggestion.

L468. One of the measured parameters was the concentration of 9'butanoyloxyfucoxanthin, a
 signature pigment for pelagophytes. I am just curious to know whether you checked the correlation
 between this pigment and the read relative abundances.

There is a weak but significant Pearson correlation between *P. calceolata* relative abundance and
 19'butanoyloxyfucoxanthin concentration in the 0.8 - 5 µm and 0.8 - 2000 µm size fractions,
 respectively 0.22 (p-value=0.02) and 0.41 (p-value=4.82e-05). In both size-fractions, the
 19'butanoyloxyfucoxanthin concentration increases with *P. calceolata* abundance, until it reaches a
 plateau. We added a sentence line 182-184.

L474. In Table 3, please add the same analysis for the 0.2-3 µm size fraction.

The 0.2 – 3 µm size-fraction shows the same tendency for the temperature and iron but is not
 significant for the iron (table below). This is probably because this size-fraction do not contains the
 entire population of *Pelagomonas* cells. Indeed, *Pelagomonas* cells diameter are around 3 µm so a
 fraction of cells may have been retained in the 3 µm filter. We prefer to present only the size-fractions
 containing all *Pelagomonas* cells in Table 3 to avoid confusion for the reader. We also removed these
 samples from Figure S6.

0.2 - 3 µm	GAM model			GAM verification		Pearson correlations	
	edf	F value	p-value	k-index	k p-value	r	p-value
s(Temperature)	2.383	2.791	0.0435	0.89	0.075	0.33	0.0008
s(Iron concentration)	1	0.005	0.9	1.03	0.5	-0.23	0.02
Adjusted R ²	0.12						
Deviance explained	15.3%						

L476. I would say, “The higher relative abundance...” instead of “The high...”. To state that, it would
 be helpful to specify which value of iron concentration generally determines iron-poor waters and
 give average relative abundances below and above this value. Support this with a statistic test
 (Mann-Whitney test, for example).

We changed the sentence according to the reviewer comment (line 179-182): *The relative*
 *abundance of P. calceolata is significantly higher in low-iron waters (203 samples <0.2 nmol/l) with*

*on average 1.77% of metagenomic reads than in high-iron environments (141 samples > 0.2 nmol/l)*
*with on average 1.11% of metagenomic reads (Wilcoxon test, p-value=1.12e-8).*

-Discussion:

L543, Include number for qPCR estimates

*Information added in the text line 294.*

Reviewer #3 (Remarks to the Author):

The manuscript by Guérin et al. describes generation, analysis, and utilization of the genome
sequence of *Pelagomonas calceolate*, a widely distributed photosynthetic marine picoplankton. In
the first part of the manuscript, the authors present details of the sequencing approach and genomic
structure and provide some high-level analysis of genome content. The second part of the
manuscript maps publicly available metagenomic sequences to the *P. calceolate* genome and uses
this information to estimate current abundance distributions and to predict distributions into the
future. In its current state, the manuscript reads like two papers with a tenuous link between the
two. For example, it is not clear how the genome structure information (introner element
distributions, GC distribution across chromosomes, impact of presence/absence of meiosis genes)
from the first half of the paper are related to the second half of the paper that focuses on *P.*
*calceolate* distributions in current and future oceans. The end result is that both halves of the paper
could be more fully analyzed. Below, I highlight areas for improvement.

*We agree that several information provided about the genome structure and content are not directly*
*related to our conclusions on the distribution in the oceans. However, we believe that it's important*
*to describe globally a genome before using it for the first time. Several genomic specificities discovered*
*in *P. calceolata* could be interesting for communications Biology audience, therefore it would be*
*inappropriate to remove these results.*

*To improve the clarity of the manuscript we focused the main text on the distribution of *P. calceolata**
*in the oceans and we moved several paragraphs about the genome structure in Supplementary Notes:*

*-Duplicated regions: Supplementary Note 1*

*-Novel Hi-C results: Supplementary Note 2*

*-Introner elements description: Supplementary Note 3*

*-GC content: Supplementary Note 4*

*-Meiosis and recombination: Supplementary Note 5*

*In addition, we changed the organization of the manuscript with the presentation of the abundance*
*based on the genome followed by the gene content and their relative expression in the environment.*

*We slightly changed the abstract and the last paragraph of the introduction to be coherent with the*
*new organization of the manuscript.*

1) The introduction consists of 5 paragraphs on changing ocean conditions, a single paragraph on
*Pelagomonas* and a single paragraph on what the authors focused on in the manuscript. There is
essentially no rationale presented for why a chromosome-resolved genome is required for
understanding the distribution of *P. calceolate* in global oceans. What does the complete genome
provide that is not provided by transcriptome or metatranscriptome data?

*A telomere-to-telomere assembly is indeed not directly necessary to study the distribution of*
**Pelagomonas* but this is the proof that our genome is complete and provide additional genomic*

information not accessible from a fragmented assembly (centromere structure, telomere repeats,
chromosome size, ...)

Transcriptomic data are less good to estimate the relative abundance of an organism because the
global transcriptomic activity varies strongly between species and according to the environmental
conditions. In addition, bacterial contaminations are frequent within a transcriptome and it's difficult
to know if a transcript homologous to a bacteria is a contamination or a recent horizontal transfer.

A complete genome gives access to the low and not expressed gene sequences in contrast to a
transcriptome where only the most expressed genes are assembled and can be studied. Finally,
genomic repeats are resolved in this nanopore assembly therefore, genes in multiple copies are
separated which is important in the interpretation of gene expression levels.

2) Methods.

346 A) The assembly step could benefit from more detail than “we selected the best assembly (Flye with
347 all reads) based on the cumulative size and contiguity.” I do not see any indication of contiguity in
Table S1. Do the different assemblies rely on default settings? They each result in vastly different
numbers of contigs. A brief description of the evidence that the longest contigs are the correct
assemblies rather than a merger of different chromosomes. A prediction of 6 chromosomes is and
‘outlier’ relative to the chromosome complement of other sequenced picoplankton as shown in
Table 1. An independent verification of aspects of the assembly should be included, such as some
PCR-based assessments in different regions.

“Contiguity” is reflected by the N50/N90 values but we agree that this term is ambiguous. We modified
this sentence. In addition, we provided details for the selection of the best assembly lines 456-460 and
in Table S1: *After the assembly phase, we selected the best assembly (Flye with all reads) based on the*
*cumulative size and fragmentation. Indeed, the Wtdbg2 and Smartdenovo assembler generated*
*fragmented assemblies with lower N90. Raven and Flye were very close but only the Flye assembly with*
*all ONT reads contained both the mitochondrial and chloroplastic circular contigs.*

To confirm the assembly performed with Flye we sequenced the same *P. calceolata* strain with the
Chromosome Conformation Capture (Hi-C) technology. We observed strong contacts within contigs
and very few across contigs. This independent verification confirmed the presence of 6 chromosomes
in the *P. calceolata* nuclear genome (new Figure S2). This new analysis is presented in the Result
section, Supplementary Note 2 and in the Mat&Met lines 402-416.

B) The authors should present the results of their BUSCO-based analysis to assess genome
completeness.

We added in the text the BUSCO completeness (94%) in the result line 119-122. The BUSCO result
details are in the new Table S4.

C) The description of the models should be expanded, particularly how model performance was
evaluated. There is relatively little information provided for a major component of the manuscript,
which is predicting relative abundance in 2099.

We provided more information on the model performance in the result section lines 198-208: *We used*
*four machine-learning techniques: Generalized Additive Models (GAM), Neural Networks (nn), Random*
*Forest (rf) and Gradient Boosted Trees (bt) and evaluated their performances with 2 parameters. The*
*Pearson correlation coefficient indicates the correlation between the model and in situ measurements*
*of P. calceolata abundance. The four machine learning tools have similar performances based on*
*Pearson’s correlations (nn =0.676; gam=0.621; bt=0.683; rf=0.694). The second parameter is the root*
*mean square error (rmse) and reflects the magnitude of the errors in the models (the number of*
*standard deviations from the mean). Using this metric the GAM approach is less good (rmse=1.04) than*

*the three other tools (nn=0.964; bt= 0.952; rf=0.941). These results indicate that we have enough in*
*situ data to capture the global trends on the relative abundance of P. calceolata but these models could*
*be imprecise on the amplitude of abundance variations. In addition, the models in the tropical waters*
*is uncertain because this environment in 2099 is out of the range of the training dataset. Because the*
*performance of the four models are similar, we combined them to obtain the most accurate projection*
*(Figure 3c and Figure S7). Despite these limitations, we projected an increase of up to 1.12% of P.*
*calceolata relative abundance from latitude 40° to latitude 50° in the North and South hemispheres and*
*a decrease in temperate and tropical waters (-0.8% maximum).*
*We also added missing details in the Mat&Met section lines 583-592.*

3) Lines 312-3313. The first sentence of the results section illustrates the limited rationale the authors
present for these studies. “To investigate its gene repertoire and its distribution across the oceans
we sequenced and assembled the genome of P. calceolata using ONT long-reads and Illumina short-
reads.” The authors then document details of how the sequence was generated.

*We changed this sentence accordingly to the new organization of the manuscript line 103: To measure*
*the abundance of P. calceolata in the oceans and study its genetic capacity to grow in different*
*environmental conditions, we sequenced and assembled the genome of P. calceolata RCC100 using*
*long-reads of Oxford Nanopore Technologies (ONT) and Illumina short-reads.*

4) Lines: 330-331. The authors present the predicted number of genes in the P. calceolate genome. In
Fig. S4, they also present the number of P. calceolate proteins homologous to proteins from other
stramenopile genomes. The authors should also provide the results of a comparison between genes
identified via transcriptome assemblies (from both the MMETSP dataset and their own
transcriptomes) and genes identified via their gene prediction algorithms. For example, they should
provide a table that includes basic stats such as the number transcriptome-assembled genes
detected in the genome assembly, the number of transcriptome-assembled genes not detected in
the genome assembly, the number of genome-assembled genes not detected in the transcriptome
assemblies. The latter number is important as a way of further justifying the power of a complete
genome assembly rather than transcriptomes from a variety of conditions. For example, were the
“meiosis-specific” genes detected only in the genome assembly and not the transcriptome assembly?

*The reviewer suggestion represents an important amount of work for a question not addressed in this*
*manuscript. Our objective is not to prove that a genome is better than a transcriptome. Our objective*
*is to present what we discovered with this chromosome-assembled genome. Even if the same*
*conclusions would have been obtained with a transcriptome, we did it with a genome, which is at worst*
*as good as with a transcriptome.*

*Yet, we rewrote 2 sentences from the discussion and the conclusion suggesting that our result can*
*only be obtained with a genome (lines 282-284/358-360/).*

5) The results on the predicted gene composition of the low-GC regions are interesting.

6) Line 381. The authors hypothesize that P. calceolate can undergo meiosis based on the genome
complement and genome structure. The authors should discuss whether there is transcriptome
support for any of the meiosis genes and if so, under what conditions. For example, are there
transcripts associated with any of these genes under exponential growth conditions and if so, what
are the implications?

*We mapped the RNAseq reads extracted during P. calceolata exponential growth and used predict the*
*genes on the genome (see Mat&Met lines 537-541). Four of the 18 meiosis genes are not expressed*
*including the double-strand DNA break initiator SPO11, suggesting that P. calceolata don't perform*

meiosis in exponential growth in lab conditions. This result was included in Table S7 and in
Supplementary Note 5. In contrast, these 18 genes are expressed in many samples in *Tara*
metatranscriptomic datasets suggesting that these genes are frequently active in natural conditions.

7) Fig. 2. The authors compared the number of nitrogen utilization genes detected in *P. calceolata* vs
the average number of these genes in seven other picophytoplankton and used the comparison to
identify genes as “overrepresented,” “equally represented,” or “absent.” The comparison to the
average is misleading. Instead, the authors should compare to the range of values found in the other
organisms.

Only two genes highlighted in their nitrogen analysis are outside the range of values found in other
organisms: glutamate synthase is present in 4 copies, which is outside the range of 1-3 in other
organisms and nitrate reductase is present in 1 copy, which is outside the range of 2-5 in other
organisms. It is a stretch to conclude from a comparison to the mean that “Among them, 8 genes
encode nitrate/nitrite or formate/nitrite transporters, which is on average higher than in other algae.
In contrast, only 5 genes encode ammonium transporters which is low compared to other species
suggesting that nitrite and/or nitrate is the main external source of inorganic nitrogen for *P.*
*calceolata*.” At a minimum, the authors should discuss the biogeography implications if these
organisms preferentially use nitrate/nitrite relative to ammonium.

The range of values is also misleading because the *Emiliania* genome has a high number of genes
(>38000) compared to other algae, so a large number of genes for almost all functions. We added to
Table S9 the gene count for each species and changed the mean by the median which is more
representative of the data. We also separated formate/nitrite from nitrate/nitrite transporters in Table
9. In the light of this new analysis and the expression data we revised our conclusions on the gene
content in *P. calceolata*. The number of nitrite, nitrate and ammonium transporter genes are not
significantly different compared other small algae so we can not conclude about the preferred source
of inorganic nitrogen. However, compared to the coastal Pelagophyceae *Aureococcus*, the number of
ammonium transporter is lower and the number of NIT-sensing genes is higher.

8) Lines 412 -414. The authors should clarify the conclusion that “*P. calceolata* is not particularly
adapted to recycle nitrogen from organic molecules but could be capable of incorporating inorganic
nitrogen compounds even in N-poor environments.” Do the authors mean to imply that there is
sufficient nitrate/nitrite in N-poor environments to support growth?

We revised this conclusion after our analysis on gene expression. If the gene content and expression
levels strongly suggest that *P. calceolata* is optimised to incorporate nitrate even in N-poor
environments, we also identified a cyanate lyase that could participate in the *P. calceolata* growth in
N-limited conditions. We added a discussion on this point line 333-338 : *Organic nitrogen compounds*
*could also be a major source of nitrogen for P. calceolata. We have shown that the cyanate lyase and*
*urease genes are expressed in many environments but only the cyanate lyase is overexpressed in low-*
*nitrate conditions. These two genes, largely present among phytoplankton lineages, could be major*
*component of acclimation to low-nitrate environments*⁵²

9) Line 429-430. The authors state “the absence of these genes [FTR1, ISIP1, ferritin] suggests that
iron uptake and storage is not a major asset of *P. calceolata* compared to the other photosynthetic
protists.” This speculative statement should be backed up by physiological data. There are many
picophytoplankton without ferritin that nonetheless can grow under a wide range of iron
concentrations.

We agree that this sentence is too speculative with our data. We removed this conclusion.

 10) The comparison of metabarcoding vs genome mapping to estimate relative abundance is
 interesting. Are there any differences if the mapping is against the available transcriptomes?

We are not sure if the suggestion is to map metagenomic or metatranscriptomic reads on *P. calceolata*
 transcriptome but in both cases, the result will be a less good proxy for *P. calceolata* abundance.
 If we align metagenomic reads on a transcriptome, reads covering introns and intergenic regions will
 not be aligned, so the relative abundance will be strongly underestimate. If we map
 metatranscriptomic reads, the number of reads will reflect the number of cells plus the transcriptomic
 activity of the population. These two analyses are possible but require large computation time (re-
 mapping of all Tara reads on several transcriptomes) and we do not think the result will be
 interpretable.

11) Lines 465 -470. It is surprising to me that PAR is not related to *P. calceolata* abundance, especially
 given that the distribution data that suggests a greater relative abundance at the DCM, particularly in
 the Indian Ocean.

PAR was not included in our analysis. We added this parameter in the revised version of the
 manuscript. The PAR (30 days average) is indeed significantly anti-correlated to *P. calceolata*
 abundance (Pearson -0.32). We included PAR measurements in the GAM model and it increases the
 explained variance of the model for the 0.8 - 5 μm size fraction (from 17.9% to 32.3%, see below).
 In contrast, we did not observe significant Pearson correlation between *P. calceolata* abundance and
 PAR in the 0.8 - 2000 μm size fraction. This is probably because there is no sample in the Mediterranean
 sea, Red sea and Indian Ocean for this size-fraction. Line 185-188

A	GAM model			GAM verification		Pearson correlations	
	edf	F value	p-value	k-index	k p-value	r	p-value
0.8 - 5 μm							
s(Temperature)	1	22.16	6.84e-06	1.11	0.87	0.23	0.001
s(iron concentration)	1.257	13.12	0.000226	0.93	0.12	-0.25	0.001
s(PAR 30 days)	1.859	15.94	4.56e-07	1.01	0.44	-0.32	0.001
Adjusted R ²	0.3						
Deviance explained	32.3%						
B	GAM model			GAM verification		Pearson correlations	
	edf	F value	p-value	k-index	k p-value	r	p-value
0.8 - 2000 μm							

s(Temperature)	3.628	9.442	1.71e-06	0.96	0.31	0.57	0.0001
s(Iron concentration)	1.540	1.225	0.24166	0.91	0.14	-0.47	0.0001
s(PAR 30 days)	2.454	6.962	0.00027	1.14	0.93	-0.051	0.6
Adjusted R ²	0.53						
Deviance explained	56.8%						

Table 2: Environmental parameters explaining *P. calceolata* relative abundance for the (A) 0.8-5 μm and the (B) 0.8-2000 μm size-fractions.

 In addition, the authors infer that nitrate concentrations are important for growth of *P. calceolata*.
 These concentrations are expected to be higher at the DCM.
 Among *Tara Oceans* stations, *in situ* nitrate measurements do not show higher concentrations at the
 DCM than the surface (see figure below and Table S10) suggesting that the greater relative
 *Pelagomonas* abundance at the DCM is not due to higher concentration of nitrate. However, we
 identified many genes in the *P. calceolata* genome involved in nitrogen compounds uptake and
 metabolism and some of them are regulated according to the concentration of nitrate suggesting that
 *P. calceolata* survival under low-nitrogen conditions is due to specific gene content and their
 transcriptomic regulation.

12) Lines 477-478. The authors should also clarify their conclusion based on model results that “the
high relative abundance of *P. calceolate* in iron-poor waters suggests that this species is particularly
capable of acclimation to this environmental condition.” Lines 429-430 (see comment #8) suggest
that “iron uptake is not a major asset of *P. calceolate*...”

Iron uptake is possible in *P. calceolata*, even though genes coding for iron chelators and ferritin are
absent from its genome, and gene coding for passive iron transporters are under- or equally
represented when compared to other PPEs. We believe that the presence of non-ferrous alternative
proteins for several key biological functions are more responsible for the acclimation of *P. calceolata*
in low-iron conditions than optimized iron transporters. We modified the sentence lines 324-327 to
precise this point.

13) Line 593-595. The authors should temper their conclusion “the chromosome- scale genome
sequence, mostly telomere-to-telomere, generated in this study is an essential starting point for its
detection in environmental datasets.” The authors have not convinced me that the telomere-to-
telomere genome sequence is required to detect organisms in environmental datasets.

We agree with the referee that abundance estimation can be obtained with a more fragmented
genome. We reworded this conclusion: *We used the chromosome-scale genome sequence, mostly*
*telomere-to-telomere, generated in this study to estimate its abundance in environmental datasets.*

REVIEWERS' COMMENTS:

Reviewer #1 (Remarks to the Author):

I am pleased that the authors have addressed most of my previous comments. One issue remained unresponded to, however. That is some effort to prove that the genes, particularly when they are more common in bacteria than in eukaryotes (e.g. TIN-sensing), were not from contaminating bacteria. Some characteristics such as GC content or phylogenetic placement can be used to address the issue.

I have one more minor comment. In line 320, "explained by" is better replaced by "is consistent with", because there are for sure many factors that make a species coastal and another oceanic.

With that, I commend the authors for their excellent work.

Reviewer #2 (Remarks to the Author):

Authors have properly addressed main reviewers concerns in the current version of the manuscript. The manuscript has been greatly improved as a result of peer review, with the addition of the metatranscriptomic data and clarifications provided for the calculation of relative abundances estimates based on metabarcoding and metagenomic datasets. I will be very happy to see this paper out!

Still there are few minor things to fix before the paper can be published:

Line 47:

Please add reference for this sentence: "Present in all oceans, PPEs are the dominant primary producers in warm and oligotrophic regions".

Lines 158-160:

Please add the number of metagenomic samples for each size fraction and depth.

Figure 3, and section "High relative abundance of *P. calceolata* in temperate, low-light and iron-poor regions" (L172). Please:

-Indicate that *Pelagomonas* abundances are based on metagenomic reads.

-Clarify which size fractions were used to represent Figure 3a (PCA). If both figures, 3a and 3b, are based on the 0.8-5 μ m size fraction, stated it clearly.

-Line 181-184. Please indicate if both size fractions (0.8-5 and 0.8-2000) were used to perform this test. The total number of samples for low (n= 203) and high (n=141) iron concentrations are not in agreement with the total number of samples stated in methods (L556) for both size fractions (0.8-2000 μ m, 119 samples and 0.8-5 μ m, 148 samples).

I recommend this entire section (Figure 3a,b and statistical tests presented on the main text) to be presented using the same set of data (not selecting one of the size fractions or both depending on the analysis); this is not clear in the reviewed manuscript.

Lines 301-302: "However, we cannot exclude that the large genome size of *P. calceolata* compared to bacterial genomes present in this size-fraction overestimates its relative abundance in metagenomic datasets".

Please indicate which size fraction are you referring to.

Line 457:

Please explain which Phred was used to determine "highest quality-scored reads".

Reviewer #3 (Remarks to the Author):

The revised manuscript by Guérin et al. is greatly improved and the authors addressed my previous comments. A few suggestions to consider for final edits

1) Figure 3 legend. There are 2 panels of PCA, but only 1 panel is described in the legend.

2) Lines 574-576 and again on line 602-604. The authors define "low-iron" as less than 0.2 nM of iron, "low-nitrate" as less than 2 μ M of nitrate and "low ammonium" as less than 25 nM. The authors should provide a reference for how they defined these values. They could also consider using the ratio of nitrate to Fe, a value expected to increase under Fe-limiting conditions, rather than individual concentrations.

Response to referees

We are thankful to the referees for their interesting comments and valuable suggestions. All modifications are in blue in this document and in track change in the manuscript file.

Reviewer #1 (Remarks to the Author):

I am pleased that the authors have addressed most of my previous comments. One issue remained unresponded to, however. That is some effort to prove that the genes, particularly when they are more common in bacteria than in eukaryotes (e.g. TIN-sensing), were not from contaminating bacteria. Some characteristics such as GC content or phylogenetic placement can be used to address the issue.

NIT-sensing genes (IPR013587 domain) are mainly found in bacteria but are also identified in several eukaryotes.

The average GC content of all *P. calceolata* genes is 64.72%. The GC content of the 3 NIT-sensing genes are 51.71% (Pca_6p09320), 69.34% (Pca_2p03520) and 61.28% (Pca_1p31590). Pca_6p09320 is located in the centromere region of contig 6 explaining its low GC content. The GC content of the two other NIT-sensing genes is similar to the average GC content in the genome, suggesting that the NIT-sensing genes belong to *P. calceolata* genome.

As described line 543: "*homologs of the 3 P. calceolata NIT-sensing genes were retrieved with a BLASTP (e-value < 10⁻⁵, coverage > 100 aa) against 27.7 million proteins from NR, the METdb79 transcriptome database, eukaryotic algal proteomes from the JGI database, Tara Oceans single-cell amplified genomes and metagenome assembled genomes (SMAGs)*". Only eukaryotic homologs were found in this BLASTP search. The phylogenetic tree presented Figure S11 shows that the 3 NIT-sensing genes have an eukaryotic origin.

In addition, we checked neighboring genes of the 3 NIT-sensing genes along *P. calceolata* chromosomes. Almost all genes surrounding NIT-sensing genes have eukaryotic homologs (best match of a BLASTP search against NR). Only the upstream gene of Pca_1p3159 has a weak match with a Verrucomicrobiales (45.54% identity)

For these three reasons we believe that the hypothesis of a bacterial contamination of NIT-sensing genes is unlikely.

I have one more minor comment. In line 320, "explained by" is better replaced by "is consistent with", because there are for sure many factors that make a species coastal and another oceanic.

We modified the sentence accordingly.

With that, I commend the authors for their excellent work.

-Senjie Lin

Reviewer #2 (Remarks to the Author):

Authors have properly addressed main reviewers concerns in the current version of the manuscript. The manuscript has been greatly improved as a result of peer review, with the addition of the metatranscriptomic data and clarifications provided for the calculation of relative abundances estimates based on metabarcoding and metagenomic datasets. I will be very happy to see this paper out!

Still there are few minor things to fix before the paper can be published:

Line 47:

Please add reference for this sentence: “Present in all oceans, PPEs are the dominant primary producers in warm and oligotrophic regions”.

We added a reference to this sentence. Agawin et al 2000, Nutrient and temperature control of the contribution of picoplankton to phytoplankton biomass and production. *Limnology and Oceanography* **45**, 591–600 (2000).

Lines 158-160:

Please add the number of metagenomic samples for each size fraction and depth.

We added these numbers in the text: *For the 0.8-5 μm size fraction, the percentage of sequenced reads aligned on the genome is 1.39% ($n=93$, $sd=1.5$) in surface samples and 2.67% ($n=55$, $sd=1.6$) in DCM samples. In the 0.8-2000 μm size fraction, *P. calceolata* represents 1.01% ($n=80$, $sd=1.2$) of all reads in surface samples and 1.56% ($n=39$, $sd=1.3$) in DCM samples.*

Figure 3, and section "High relative abundance of *P. calceolata* in temperate, low-light and iron-poor regions" (L172). Please:

-Indicate that *Pelagomonas* abundances are based on metagenomic reads.

We added “metagenomic-based” in the legend of figure 3 and in the sentence line 175: *Principal component analysis revealed a positive relation between metagenomic-based *P. calceolata* abundance, the temperature and the coast distance and a negative relation with iron concentration (Figure 3a and b).*

-Clarify which size fractions were used to represent Figure 3a (PCA). If both figures, 3a and 3b, are based on the 0.8-5 μm size fraction, stated it clearly.

We modified the figure legend to precise this information: *Principal component analysis of the metagenomic-based relative abundance of *P. calceolata* in the 0.8 - 5 μm size fraction*

-Line 181-184. Please indicate if both size fractions (0.8-5 and 0.8-2000) were used to perform this test.

This test was performed on all size fractions. To be consistent across the paragraph we did it separately for the 0.8-5 μm and the 0.8-2000 μm size-fractions. We modified this sentence: *In*

*the 0.8 - 5 μm size fraction, the relative abundance of *P. calceolata* is higher in low-iron conditions (<0.2 nmol/l, 54 samples) with on average 2.3% of metagenomic reads than in high-iron environments (>0.2 nmol/l, 88 samples) with on average 1.7% of metagenomic reads (Wilcoxon test, $p\text{-value}=0.02$). In the 0.8 - 2000 μm size fraction, we observe the same tendency with a relative abundance of 1.9% of metagenomic reads on average in low-iron waters (49 samples) and a lower relative abundance of 0.78% of metagenomic reads on average in high-iron environments (59 samples) (Wilcoxon test, $p\text{-value}=9.6e^{-7}$).*

The total number of samples for low (n= 203) and high (n=141) iron concentrations are not in agreement with the total number of samples stated in methods (L556) for both size fractions (0.8-2000 μm , 119 samples and 0.8-5 μm , 148 samples).

This incoherence was due to the presence of the 0.2-3 size fraction in the previous calculation. We modified the sentence and sample counts are now coherent (see above). We don't have iron concentrations for 17 samples explaining the slight difference between the total number of samples indicated in the Mat&Met (119+148 = 267) and the number of samples indicated in the results (88 + 54 + 49 +59 = 250).

I recommend this entire section (Figure 3a,b and statistical tests presented on the main text) to be presented using the same set of data (not selecting one of the size fractions or both depending on the analysis); this is not clear in the reviewed manuscript.

This entire section is now on the same set of data. All results are presented for the 0.8 – 5 and 0.8 – 2000 μm size-fraction separately. The PCA for the 0.8 - 2000 μm size-fraction is in Supplementary Figure S6b to reduce the number of main figures.

Lines 301-302: “However, we cannot exclude that the large genome size of *P. calceolata* compared to bacterial genomes present in this size-fraction overestimates its relative abundance in metagenomic datasets”.

Please indicate which size fraction are you referring to.

We modified the sentence: *However, we cannot exclude that the large genome size of *P. calceolata* compared to bacterial genomes present in the 0.8 - 5 μm size-fraction overestimates its relative abundance in metagenomic datasets.*

Line 457:

Please explain which Phred was used to determine “highest quality-scored reads”.

Our previous answer was incorrect. Fitlong tool was used using *Pelagomonas* illumina short reads as reference. In this case, the nanopore read selection is not based on the Phred score but on the k-mer matches to Illumina short reads. Nanopore reads well-covered by Illumina reads have higher scores. A more accurate gauge of quality according to Fitlong authors. Default parameters of Fitlong were used. We modified the sentence to clarify this point: *For the genome assembly, we generated three sets of ONT reads: all the reads, 30x genome coverage with the longest reads and 30x genome coverage of the highest quality reads estimated by the Fitlong tool (<https://github.com/rrwick/Fitlong>). We applied Fitlong with default parameters using*

Pelagomonas Illumina short reads as a reference (ONT reads covered by Illumina reads have higher scores).

Reviewer #3 (Remarks to the Author):

The revised manuscript by Guérin et al. is greatly improved and the authors addressed my previous comments. A few suggestions to consider for final edits

1) Figure 3 legend. There are 2 panels of PCA, but only 1 panel is described in the legend.

We modified the legend to describe the top and bottom panels of Figure 3a.

2) Lines 574-576 and again on line 602-604. The authors define “low-iron” as less than 0.2 nM of iron, “low-nitrate” as less than 2 μ M of nitrate and “low ammonium” as less than 25 nM. The authors should provide a reference for how they defined these values.

To our knowledge, there is no reference in the literature indicating which threshold should be used to define an environment as low-nitrate, low-iron, or low-ammonium for phytoplankton growth analysis. Therefore, we defined threshold based on the distribution of nutrient concentrations in our dataset (see figure below, the red lines are threshold used for this study). For nitrate and ammonium concentrations there are slight bimodal distribution, we choose a threshold between the 2 peaks. For iron concentration we used the threshold defined previous studies on the same dataset (Carradec et al., 2018 and Caputi et al., 2019). We modified the sentence line 577: *We consider oceanic samples as “low-iron” if they contain less than 0.2 nM of iron, “low-nitrate” if they contain less than 2 μ M of nitrate and “low-ammonium” if they contain less than 25 nM of ammonium. These thresholds were defined with the distribution of nutrient concentrations in the dataset and previous studies^{13,14}.*

They could also consider using the ratio of nitrate to Fe, a value expected to increase under Fe-limiting conditions, rather than individual concentrations.

The ratio of different nutrients is an interesting suggestion to provide more information on the

biological conditions limiting or increasing *P. calceolata* growth. However, this is complex and many ratios could be tested (N:P ; C:N,...). We choose in this study to evaluate the impact of many parameters individually to discover the most important factors. Future studies could go into more details combining different parameters.